# Drone-Aided Delivery Methods, Challenge, and the Future: A Methodological Review

**Xueping Li** *,†, **Jose Tupayachi** †, **Aliza Sharmin** † and **Madelaine Martinez Ferguson** †

Department of Industrial and Systems Engineering, University of Tennessee, Knoxville, TN 37996, USA
* Correspondence: xueping.li@utk.edu; Tel.: +1-865-974-7648
† These authors contributed equally to this work.

**Abstract:** The use of drones for package delivery, commonly known as drone delivery or unmanned aerial vehicle (UAV) delivery, has gained significant attention from academia and industries. Compared to traditional delivery methods, it provides greater flexibility, improved accessibility, increased speed and efficiency, enhanced safety, and even some environmental benefits. With the increasing interest in this technology, it is crucial for researchers and practitioners to understand the current state of the art in drone delivery. This paper aims to review the current literature on drone delivery and identify research trends, challenges, and future research directions. Specifically, the relevant literature is identified and selected using a systematic literature review approach. We then categorize the literature according to the characteristics and objectives of the problems and thoroughly analyze them based on mathematical formulations and solution techniques. We summarize key challenges and limitations associated with drone delivery from technological, safety, societal, and environmental aspects. Finally, potential research directions are identified.

**Keywords:** drones; unmanned aerial vehicle; last mile logistics; parcel delivery; mathematical programming

## 1. Introduction

The advancement of Industry 4.0 has significantly increased the capabilities of drones, which have found broad applications in defense, SAR (Search and Rescue), agriculture, industry, and logistics [1,2]. Most recently, drone application has observed a rapid surge during the global COVID-19 outbreak [3–6]. Due to the implementation of restrictive social measures for COVID-19, large retail chains and package delivery firms were compelled to seek improvements in their logistic operations [7,8]. Further, several online businesses started offering same-day delivery that further enhanced customer expectations for quick delivery [9,10]. That is why companies such as Amazon, DHL, and FedEx are trying to adopt drones for last mile delivery (LMD) for faster and more effective delivery and to generate profits [11,12].

Drones are not just limited to delivery services; they have a wide range of applications in various fields, including military, construction, security, health, precision farming, disaster management, and surveillance [13]. In the past, drones were primarily used for military operations to track enemy movements and for target killing. Nowadays, they are also used for traffic surveillance, image and video mapping, and exploring hard-to-reach areas. In agriculture, drones collect real-time data that helps farmers make informed decisions about adjusting their farming inputs to achieve better yields [14]. In healthcare, drones are used to deliver emergency medical supplies to remote areas quickly, reducing the risk of complications and fatalities. Additionally, drones are valuable tools for disaster management, enabling the efficient delivery of aid to affected areas.

Besides the interest from industries, drones have also recently gained much literary attention from academia. Mohsan et al. [13] provided a comprehensive review of drone

classification, characteristics, and various applications. Hassanalian and Abdelkefi [15] classified drones into six categories on the basis of their configuration and discussed their applications under three types: mission, flight zone, and environment. Further, the study surveyed drone design challenges, as well as manufacturing methods to overcome those challenges. Several other pieces of literature have focused on the current and potential future applications of drones [14,16,17]. Rejeb et al. [2] explored capabilities, challenges, and potential outcomes of drones particularly employed in humanitarian logistics and management. Benarbia and Kyamakya [18] provided an extensive survey on the development of drone delivery systems, along with feasibility and performance, while suggesting some future research directions. Büyüközkan and Ilıcak [19] reviewed smart urban logistics and identified the most common technologies explored as well as the related technologies with the potential to be implemented. Mohd Noor et al. [20] explored the challenges of drone application in urban settings, emphasizing the efficiency of drones in solving urban issues while ensuring a sustainable and efficient environment. Kellermann et al. [21] analyzed possible barriers and problems that can impede the acceptance of drones as a medium for transporting parcels and passengers. It also discussed the expected societal benefits of drone usage and emphasized evidence-based promises to gain a social acceptance of drones as a transportation medium. Khoufi et al. [22] investigated the existing literature on drone path optimization, focusing specifically on the Traveling Salesman Problem (TSP) and Vehicle Routing Problem (VRP). However, fewer mathematical modeling details of these problems were provided. Chung et al. [23] reviewed the literature on various optimization issues related to drone-only and drone-truck combined operations in depth, including drone routing, drone scheduling, task assignment, area coverage, and communication. The study discussed various mathematical modeling techniques, drone truck synchronization, and ways to overcome its barriers in detail. Rojas Viloria et al. [24] focused on the challenges with drone routing and classified the literature based on application, objectives, solution approaches, and constraints considered. They grouped the literature into five sections: internal logistics, entertainment, parcel delivery, military, and surveillance and data collection. Moshref-Javadi and Winkenbach [25] also classified the drone logistics literature on the basis of applications, the same as Rojas Viloria et al. [24], but they explicitly focused on e-commerce, healthcare, postal services, food delivery, and emergency services. Further, the authors divided the logistics models of drones into four broad sections: drone-based pure-play, unsynchronized, synchronized, and resupply multi-modal models. Rejeb et al. [26] discussed the possible benefits of drone deployment in supply chain management and identified the challenges in the real world. Macrina et al. [27] reviewed drone-aided routing with a focus on parcel delivery. They classified the delivery routing problems into four parts: drone-based TSP, drone-based VRP, drone delivery problem, and carrier vehicle problem with drones, with literature from 2015 to 2020.

In this review, we focus on the most recent development of drone-aided delivery research and cover the papers published between 2015 and 2022, from a methodological aspect that could benefit both researchers and practitioners. Specifically, to the best of our knowledge, no existent reviews concentrated on drone-aided parcel delivery problems with a focus on urban area challenges. Thus, this paper provides a centered and updated review that classifies and thoroughly analyzes drone delivery problems covering various mathematical algorithms as well as potential challenges.

The remainder of the paper is structured as follows: Section 2 discusses the approach implemented for this systematic review. Section 3 provides a detailed examination of the drone delivery literature, which is divided into five sections. Section 4 examines the limitations of deploying drones for distribution in urban environments. Finally, Section 5 concludes the paper.

## 2. Review Methodology

The increasing popularity of consumerism has led to a growing interest in drone-assisted delivery, particularly in urban areas. However, there is a lack of research specif-

ically analyzing the use of drones in delivery operations and the challenges they face in urban environments. The goal of this review is to fill this gap by providing a comprehensive examination of drone-aided delivery, highlighting current issues, and identifying potential areas for future research. To achieve this, a systematic approach is adopted, which is commonly used in medical, management, and supply chain literature [26,28]. This approach involves several structured steps [29], such as identifying research gaps, conducting a thorough literature review using specific criteria and keywords, selecting relevant literature, classifying the literature for a better understanding, synthesizing the information, and ultimately providing findings and recommendations for future research. A detailed overview of the research methodology is demonstrated in Figure 1.

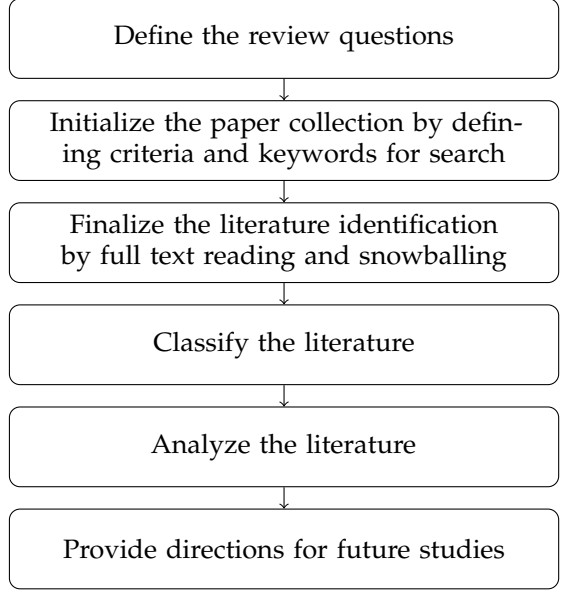

**Figure 1.** Detailed research methodology.

### 2.1. Research Question Formulation

Three main research questions are formulated for this study on the basis of the gaps identified in the existing literature. The formulated questions are as follows:

i.   What are the research methods used for drone-aided delivery in the existing literature?
ii.  What are the problems or challenges in drone-aided delivery in urban areas and what solutions have been implemented so far?
iii. What are the possible future research directions for advancing drone-aided delivery?

### 2.2. Literature Collection

The search process is initiated by locating and identifying all relevant studies that align with the stated review agenda. The Scopus database is utilized to perform a relevant literature search on the basis of the selected keywords in September 2022. Scopus is well-recognized for being effective, accurate, and comprehensive, and it covers wide-ranging and diverse literature areas in technology, science, medicine, art, humanities, and social sciences [30]. To accumulate the relevant literature for review, the following string (set of keywords) was developed and utilized for the database search:

(Drone*) OR ("Unmanned aerial vehicle*") OR (UAV*) AND ("last mile delivery*") OR ("parcel delivery*") OR (delivery*) OR (logistics) OR (routing) AND (urban*).

The search compiled the literature consisting of these keywords in the title, abstract, and keywords fields. Different acronyms or terms were used in this string for addressing drones to include all the literature relevant to our questions, regardless of what the existing literature has chosen to define as drones. To account for the multidisciplinary nature of the

research regarding drones, the search was not restricted by any filter, as it could limit the results. The search result comprises all international English language publications that appeared in the Scopus database from the year 2015 to September 2022, yielding 165 articles, conference papers, or book chapters. A spreadsheet was created for the ease of article tabulation, facilitating their systematic inspection. Figures 2 and 3 represent a distribution of papers based on the year and country, accordingly.

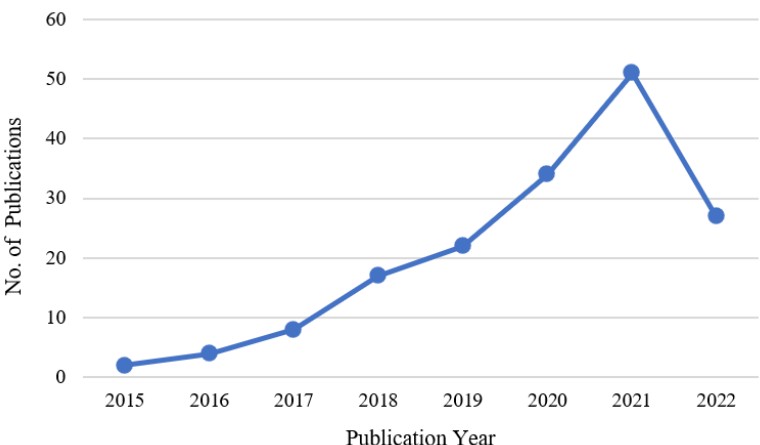

**Figure 2.** Year-wise distribution of retrieved publications.

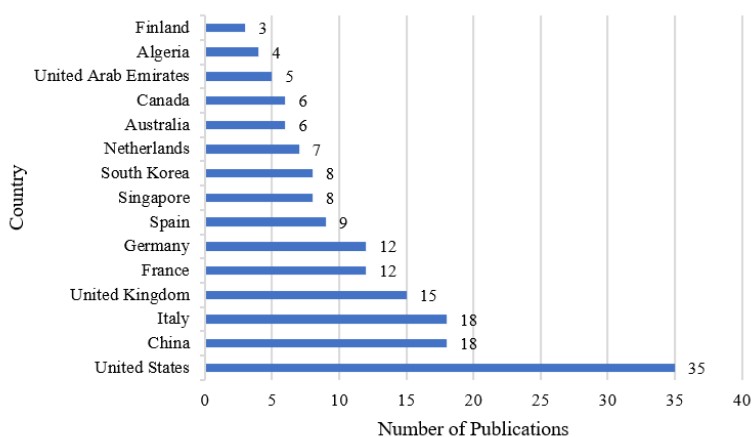

**Figure 3.** Country-wise distribution of retrieved publications.

*2.3. Literature Selection and Classification*

The assessment and selection of articles were executed by screening and scrutinizing the articles in three steps. In the first screening, the title and abstract of the articles were given a full read. It was found that although the keywords were present in some of the articles, they were not the main topic of the article, or the article did not explicitly focus on drone delivery or urban drone delivery, or both; thus, they were eliminated. Articles that did not have the full text available online were also excluded: 77 articles out of 165 were selected in this step. In the next round, the potential articles were screened by the full-text reading and were excluded if not relevant to the scope of our review. In this round, 13 articles were excluded. Finally, additional relevant articles were identified through a snowball sampling of references of the selected articles. Backward snowballing was utilized to find which references the start set of the articles cited to build their case. The papers were assessed on the basis of the title, keyword, abstract, and how and where they were cited. This three-step process ensured the alignment of the scope between the final articles and our research agenda. Following this process, a final data set of publications for the review and analysis consisting of 73 articles were obtained, including the articles from the snowballing

method. Further, the extant literature is classified into five groups, namely: the traveling salesman problem, vehicle routing problem, drone delivery scheduling problems, drone optimization problems, and urban last mile problems.

## 3. Literature Analysis

### 3.1. Traveling Salesman Problem (TSP)

TSP is a mathematical problem that attempts to determine the shortest path, given that a certain node is only covered once. Figure 4 illustrates the problem in which the top 50 most populous US continental cities are the customers, and the salesman visits each customer only once and returns to the depot (the city of Memphis). This is one of the most popular problems in the existing drone delivery literature. Publications addressing TSP problems are divided into two main categories: collaboration and parallel work. Then, they are grouped by the applied method and explored in detail.

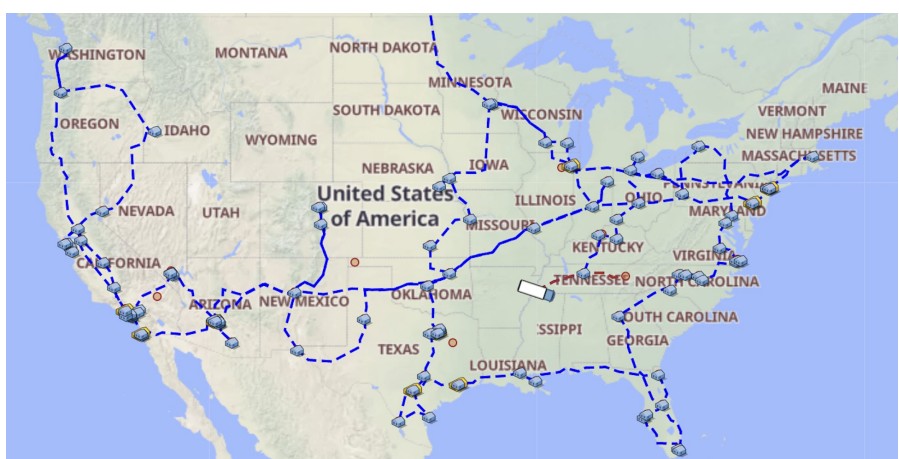

**Figure 4.** Traveling Salesman Problem—an illustration. A truck traverses the top 50 most populous US cities starting from Memphis.

### 3.1.1. Exact Method-Based Approach

Exact methods aim at obtaining an optimal solution and do not rely on heuristics. In this section, a cooperative technique underpins the operation of the last mile delivery employing drones and vehicles. Yurek and Ozmutlu [31] formulated the problem based on the concept proposed by Murray and Chu [32] to build a two-level decomposition to systematically tackle a tandem delivery. In this study, 12 instances were solved in 15 min. The decomposition method involved covering the drone's path after the truck tour is defined. The optimization tasks followed a sequence based on node satisfaction, rendezvous point, and tandem or parallel travel. Cavani et al. [33] constructed a mixed integer linear program (MILP) formulation to reckon a tandem-based single-truck and multiple drones. A branch-and-cut implementation addressed the formulation via an exact decomposition and a compact method leading to an optimally proven solution during a two-hour computing time employing three drones. Boccia et al. [34] further expanded the challenge presented by Murray and Chu [32], except for allowing the vehicle to wait at the launch node. The suggested approach is based on a sub-tour split based on the max-flow min-cut (MFMC) approach and an integer linear programming (ILP) formulation. This allowed an optimal solution for up to 20 tested instances. Similarly, the proposed method by Boccia et al. [34] benefits from the removal of the big-M constraints for vehicle synchronization and carries a procedure of a sequential selective introduction of path variables that generate an improvement of 4.10% over the base model. Kim and Moon [35] used a mixed ILP (MILP) formulation that is comparable to Cavani et al. [33] with the inclusion of drone stations, which serves to store, charge, and relaunch. The formulation is decomposed into an independent TSP and a parallel scheduling issue. In this sense, it is specified that the vehicle station supplier operates independently of the tandem delivery. This formulation is

operated as both a TSP and a parallel identical scheduling issue. Finally, Bouman et al. [36] discussed a strategy based on dynamic programming; specifically, it is an exact technique known as the Bellman–Held–Karp dynamic programming algorithm. The method enumerates the shortest paths and identifies the least expensive travels, allowing one to solve numerous node instances.

### 3.1.2. Heuristics

Heuristic methods run fast and can produce feasible and near-optimal solutions if constructed properly. In such a regard, heuristics meet real-world industry needs by providing deployable fast solutions. A MILP-based heuristic method is proposed by Murray and Chu [32], in which only a node or vehicle is designated as the drone's landing location. As the authors pointed out that the MILP formulation is NP-hard, the Clarke–Wright savings heuristics are employed. Common parameters, such as the number of customers, feasible drone-delivered consumers, and speed restrictions for trucks and drones, are established in this study. The findings demonstrated that even with the inclusion of a large battery capacity, the slow travel speed restricted the number of drone-delivered packages. Additionally, Ha et al. [37] used a similar strategy to Murray and Chu [32]. However, it attempted to minimize operational expenditures. The best results are obtained through a local search and a greedy randomized adaptive search Procedure (GRASP), and the use of algorithms such as K-nearest neighbor, k-nearest insertion, and random insertion. The proposed procedure begins at the depot and visits the nearest nodes before inserting the non-visited nodes repeatedly, utilizing a sample of nodes. Each iteration is based on a two-step procedure that examines every client in order to select the best candidate for relocation. Upon a solution improvement, the truck route and sub-route are adjusted, and the client is eliminated. The GRASP algorithm is found to be better than the local search method. A comparable method is adopted by Almuhaideb et al. [4] that included two local search choices and a self-adaptive neighborhood. Marinelli et al. [38] presented a modified version of the TSP drone tandem delivery that maximized drone endurance and utilization. This heuristic is based on the GRASP method and focused on the truck-drone activities that must occur at customer nodes. While the Lin–Kernighan (LKH) heuristic is used to locate all viable truck–drone operations with reduced costs, the results demonstrate that the along-the-arc operations outperform the greedy heuristic. Hence, the higher drone speed en-route-alone benefits become less meaningful. de Freitas and Penna [39] addressed the usage of a single truck and a drone, offering a hybrid heuristic neighborhood search (HGVNS) to select truck and drone paths. It incorporated a precise, three-step technique, which is based on a TSP solution created to determine the truck delivery route to every customer. Next, truck customers are eliminated and replaced with drone customers while using a greedy technique for finalizing the HGVNS update and improving the current solution. The results yielded a 9% to 12% improvement from the base model, and the delivery time was reduced as the drone speed increased. Ha et al. [40] constructed a method that is comparable to Murray and Chu [32]. The dynamic population management and hybrid genetic search include new search operators, a penalty mechanism, and a technique for converting the TSP problem into a chromosome. The method can be summarized as the first spring generation, whose offspring are then schooled for a general update of the "tour" chromosome. It included a penalization cost to achieve a balance between intensification and diversity while avoiding early convergence. After that, the education stage is carried out using a local search approach. Then, the massive tour is updated. The suggested strategy increased the number of instances and outperformed the similar strategies covered. Lin et al. [3] addressed a similar issue with the goal of increasing the UAV profitability using the model of upcoming requests. Penalty variables for delay, early arrival, and capacity inclusion are the main focus of the authors' evolutionary algorithms. On the other hand, Kitjacharoenchai et al. [41] built a mixed integer program (MIP) formulation to reduce the arrival of drones that can land in any adjacent truck. In addition, an adaptive insertion heuristic (ADI) handled up to 100 nodes, reducing the last delivery time com-

pletion. The algorithm required drones to only combine with trucks at a client site, and only one drone could be released and collected at a time. The heuristic is based on the greedy node insertion strategy and builds an initial multiple TSP truck-only delivery in two stages. Raj and Murray [42] addressed the multiple flying sidekicks' TSP with variable drone speeds, in which a three-step heuristic, tour partitions, scheduled activities, and timings power a local search method with speed and range trade-offs present. Finally, a contribution that extended the variation of the single truck and the work of multiple drones is presented by Baniasadi et al. [43]. It is based on an Integer Programming (IP) formulation and a clustered and generalized TSP, where the LKH heuristic and cross-entropy (CE) meta-heuristic algorithms are employed.

A second stream of problems using heuristics is the parallel work collaboration between a truck and a drone. Murray and Chu [32] presented a new concept known as parallel drone scheduling. It is based on a single delivery vehicle and a number of drones that depart and return from the distribution center, which evaluates clients in close proximity to the distribution site. Plus, operations are held without any coordination between the involved vehicles; the truck serves the TSP route, and the drones serve customers in the vicinity of the distribution center. To handle the suggested minimization/maximization mission time among the vehicles, an MILP formulation, a savings-based heuristic, and the nearest neighbor algorithm are applied. Dell'Amico et al. [44] also evaluated the use of heuristics to tackle this novel problem. A "Fast heuristic" applied the LKH algorithm to generate a tour for a specific consumer in a series of nodes. Moreover, an iterative variant is covered in which the initial tour is computed before applying a classic 2-opt topological exchange in orderto finalize with a random restart local search (RRLS) updating the MILP. The fast heuristic performed faster than RRLS. The multiple warehouse delivery problem (MWDP) is covered by Mathew et al. [45]. Two methods in the book rely on enumeration and a reduction to a TSP formulation where a single drone and several static warehouses are used. First, it is built on an algorithm that converts an MWDP into a TSP and can be solved using either the LKH heuristic or exact approaches. Second, the kernel sequence enumeration (KSE) is based on enumerating a transitional delivery location between two warehouses and an ordered subset of warehouses. The approaches depicted by the MWDP produced results that were competitive; however, the KSE method's superiority over the TSP LKH proposal is decreased by the presence of more warehouses. Saleu et al. [46] extended the work in the study of a parallel drone truck delivery. The proposed method built a giant tour and then split it to determine the set of vehicles to employ. This formulation is based on a MILP. Then, hybrid meta-heuristics are adapted from an iterative two-step heuristic that visited all customers and used dynamic programming for efficient customer partitioning. Constraints' relaxation and a local search are also applied to improve the solution. Over time, the formulations have evolved significantly. Initially, they only included basic factors such as the sidekick approach. However, the formulations have become more complex and now take into account factors such as the flying capacity from the fly tandem, truck–drone meet-up operation, clustered delivery, and the inclusion of supplier stations. These additions have resulted in some constraints being updated accordingly.

A summary of the reviewed papers related to TSP can be found in Table 1.

**Table 1.** Summary of TSP-related papers.

| Author | Approach | Future Directions |
|---|---|---|
| Yurek and Ozmutlu [31] | • Iterative decomposition approach based on MIP formulation and three synchronizations' constraints | • Clustered and hybrid algorithms |
| Dell'Amico et al. [44] | • No truck sync and VRP generalization | • Larger instances of heuristics |
| Marinelli et al. [38] | • Lin Kernighan heuristic, the inclusion of en-route operations | • Dynamic simulation with en route and congested arcs |
| Saleu et al. [46] | • Giant initial tour and hybrid meta-heuristics<br>• Improvement steps and MILP formulation | • Exact solution branch-and-price<br>• Constraint Programming framework |
| Kitjacharoenchai et al. [41] | • Two-phase heuristic and adaptive insertion algorithm, initial multiple TSP<br>• Genetic algorithm, combined k-means/nearest neighbor, random cluster/tour | • Adaptive large neighborhood search<br>• Simulated annealing mimetic algorithm |
| Baniasadi et al. [43] | • IP formulation of clustered generalized TSP (CGTSP), transformed CGTSP and heuristics LKH and CE | • Fine tuning TSP heuristics<br>• Delivery nodes clustering |
| de Freitas and Penna [39] | • Optimal TSP solution through a MIP solution, implementation of the general variable neighborhood search (meta-heuristic) | • MILP formulation<br>• Multiple delivery trucks and drones |
| Ha et al. [37] | • Implementation of a min-cost TSP objective<br>• MILP model; GRASP and split-based TSP | • Meta-heuristics<br>• Multiple vehicles and multiple drones |
| Mathew et al. [45] | • TSP: LKH suboptimal solutions, noon-bean transformation<br>• LKH heuristic | • Simultaneous deliveries and drone capacity greater than 1 |
| Boccia et al. [34] | • MILP and ILP formulation<br>• Path-based formulation for vehicle sync | • Operational constraints and algorithmic refinements |
| Murray and Chu [32] | • Flying Sidekick TSP (IP, Heuristic: savings, nearest neighbor, sweep)<br>• Parallel Drone TSP: (IP, Heuristic: savings, nearest neighbor) | • Sophisticated local search, simulated annealing, and Tabu search |
| Raj and Murray [42] | • Tour partition<br>• UAVs' sorties, scheduled activities, and local search algorithm | • Heuristics, multi-track problems, and en-route operations |
| Lin et al. [3] | • GA for the global optimal solution | • Multiple rush requests and volume costs |
| Kim and Moon [35] | • TSP drone station and MIP model separation in TSP multiple stations | • Multiple drones' station |
| Cavani et al. [33] | • MILP formulation, decomposition approach, and branch-and-cut algorithm | • Multiple trucks, en-route operations, and uncertainty |
| Ha et al. [40] | • Hybrid genetic algorithm with dynamic population management and adaptive diversity control based on a split algorithm | • Multiple trucks and drones |
| Almuhaideb et al. [4] | • Greedy randomized adaptive search, two local search alternatives, and a self-adaptive neighborhood | • Neighborhood search alternatives |
| Bouman et al. [36] | • Bellman–Held–Karp dynamic programming and shortest path enumeration | • Multi-drone operations and any point departure |

### 3.2. Vehicle Routing Problem (VRP)

VRP can be considered as a generalization of TSP with multiple routes to visit all the nodes with the same starting point. Figure 5 illustrates the optimal route for a VRP problem with one depot with a fleet of capacitated vehicles and several target nodes that should be visited exactly once. Each route starts at the depot, visits a subset of nodes, and then returns to the depot. Variations of VRP have been studied in the last few years, considering drones as part of the fleet [47–57] or the unique fleet [51,58–62] to be assigned in the delivery process.

Sixteen articles from the searched results were classified in this category, with reference to the corresponding research methods and results obtained. Particularly, the relevance of these optimization methods relies on the techniques developed by the researchers considering the complexity of the VRP with drone problems, which are, in many cases, considered NP-hard

problems, as [47,52] mentioned. The most common research method found in the literature regarding the consideration of a fleet of drones in the VRP is the Graph Theory [50–52,54,59–61], followed by MIP [47–49,58]. On the other hand, the less common ones were MILP [57], IP [56], Worst-case Analysis [55], Statistical Techniques [53], and Stage-wise Modeling [62].

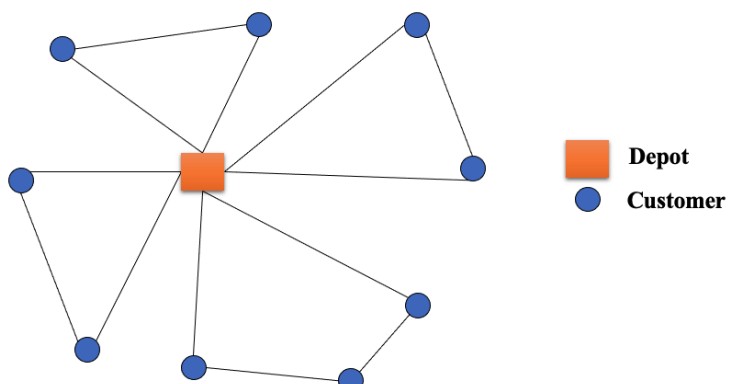

**Figure 5.** An illustration of the Vehicle Routing Problem.

Regarding Graph Theory, Daknama and Kraus [50] centered their research on the minimization of the delivery time, specifically, the average delivery time of the packages, whilst Othman et al. [52] formulated four different truck-drone scenarios related to the truck waiting time with the objective function established to find the minimum cost path. In the case of [50], the delivery is performed by any of the two vehicles, contrary to [52], where the last mile is performed exclusively by drones. To solve the problem, a heuristic approach was developed by Daknama and Kraus [50]. They used a two-nested local search by starting from a pre-initial solution (treating the problem as a TSP), then adding the drones to each truck, and finally evaluating the exchange of the packages between the trucks. Not only did Daknama and Kraus [50] apply heuristics, Othman et al. [52] also adopted this strategy but from a theoretical perspective, first demonstrating the complexity of the problem as NP-hard and proposing a polynomial-time approximation algorithm to solve it. As for Pugliese et al. [51], a comparative analysis of three transportation systems (only drones, only trucks, and a truck–drone pair) was conducted considering the constraints related to time windows and synchronization, as well as $CO_2$ emissions. The results associated with their study highlighted the efficiency of the truck-drone system over the other two systems in terms of cost, amount of customers served, and $CO_2$ emissions.

Other contributions related to Graph Theory were made by Thibbotuwawa et al. [59] and Zhu et al. [60]. The study conducted by [59] had a narrow focus on the fulfillment of the customer's demand with consideration of the time window, weather conditions, drone energy consumption, and collision avoidance. To solve the problem, Thibbotuwawa et al. [59] employed a decomposition technique to break the problem down into five smaller components, which helped them discover relationships between wind speed and drone battery capacity in order to fulfill a certain amount of customers. Like Thibbotuwawa et al. [59], Zhu et al. [60] concentrated their research on potential outside influences on drone delivery. In their specific situation, they assessed the effects of random shock in the drone's route while pursuing the goal of minimizing the overall cost influenced by the drone's destruction, package destruction, and unattended consumers. To solve the proposed problem, they first calculated an approximation of the damaged packages through a Monte Carlo Simulation and then applied a Tabu search algorithm.

In the case of Cheng et al. [61], a 2-index formulation was applied with constraints related to the drone's capacity, customer's time windows, and drone battery capacity, whereas a nonlinear model was used to examine the drone's energy usage. To solve this formulation, Cheng et al. [61] developed a branch-and-cut algorithm. Additionally, Chang and Lee [54] attempted to cluster the delivering points to define a path for a truck

within the middle point of the clusters and drone paths within each cluster. Their approach was structured in three steps, starting with the clustering through the K-means clustering technique. Then, they used TSP for the truck routing throughout the cluster's center. Finally, a nonlinear programming model is adapted to shift the cluster center closer to minimize the truck route, and, considering that the drone speed is higher, widening the drone flight range will lead to the minimization of the total delivery time.

In terms of the problem formulation, within the MIP, Sacramento et al. [47] expanded the model developed by Murray and Chu [32], which was originally for the TSP, including constraints related to the capacity and time completion, while the cost minimization was settled as the objective function. A similar idea was followed by Huang et al. [48], while Xia et al. [58] focused on a multi-drone problem, which includes multi-customers' delivery through the establishment of collaboration between the operators, a blockchain-enabled fleet sharing approach.

Even though the mathematical model is rooted in the MIP and some similarities were found in the constraints used, the algorithms developed by the researchers vary while tackling the complexity of the VRP with the drone problem. In that direction, heuristics were used by Sacramento et al. [47], Huang et al. [48], Lin et al. [49], while Xia et al. [58] were more inclined to an exact approach. Sacramento et al. [47] proposed the Adaptative Large Neighborhood Search (ALNS), a meta-heuristic algorithm, while Huang et al. [48] exploited an Ant Colony Optimization (ACO) algorithm. Both were tested with random instances resulting in significant cost savings for the drones' fleet incorporation by comparing them with the VRP. To solve a multi-drone and one truck routing problem, Lin et al. [49] decomposed the problem formulation to be solved through an h-GA and a hybrid particle swarm optimization (h-PSO). In relation to the exact algorithm method, Xia et al. [58] developed a branch-and-price-based algorithm to solve the MIP formulation. In the case of Tamke and Buscher [57], two MILP models were developed. One of them pursued a min-sum approach while the other resulted in a min-max, with the objective of minimizing the maximum fleet's completion time while the former seeks the minimization of the total fleet's completion time. Both were solved by a branch-and-cut algorithm.

In the research by Wang et al. [55], the formulation considered that the drones keep the same truck's assignment during the delivery process while pursuing the minimization of the fleet completion time through a Worst-case Analysis. The cases involved the assessment of configurations regarding the number of drones, the drone's speed in contrast with trucks, and the assignment of customers, all of them from a theoretical standpoint, the same as Othman et al. [52]. The results established that the completion time can be improved up to four times with the use of drones or up to 75% in other cases. A similar problem was studied by Luo et al. [56]. Their attention was focused on defining a routing strategy for the truck–drone pair, where the former maintains the road network while the drone makes the delivery to customers beyond the road that cannot be reached by truck. In this case, the truck acts as a mobile depot for the drone. Both vehicles concur at some points of the road with the aim to change or charge the drone's battery. Considering all of this, the research objective was to find a feasible solution due to the minimization of the drone's route, including its energy capacity. They developed and tested two constructive heuristic algorithms to solve the MIP formulation proposed as well. This algorithm resulted in a better computational time to achieve a feasible solution.

The research of Ulmer and Thomas [53] included a time window constraint to the problem, commonly known as the same-day delivery (SDD), as well. The overall objective was to maximize the amount of served customers while the delivery costs were minimized. Consequently, they used a Parametric Policy Function Approximation (PFA), using a vehicle's time limit from the depot as a parameter. The research's results indicated the benefits of the combination of both vehicles in the fleet versus the use of only one so as to geographically divide the delivery area in order to improve the "overall number of potential services significantly" ([53], p. 476). The drones' travel range limitations during the delivery process were interestingly explored by Choudhury et al. [62], in

which the authors presented an algorithm developed through a stage-wise approach. The aim of their approach was to take advantage of the existing ground transit network to shorten the drone's flight. This approach included an upper and lower layer: one involved the task allocation of drones and depots, while the other stood for the use of the existing urban transit network. In the same direction, they developed an approximately optimal polynomial for the upper layer and a multi-agent path-finding algorithm for the lower layer.

A summary of the reviewed papers related to VRP can be found in Table 2.

**Table 2.** Summary of VRP-Related Papers.

| Author | Approach | Future Directions |
|---|---|---|
| Daknama and Kraus [50] | • Graph theory<br>• Two nested local search algorithms<br>• Local search and outer local search | • Consider adding to the model packing time, time windows, charging the battery, and other constraints<br>• Consider a variation of the model with drone landing in moving vehicles |
| Othman et al. [52] | • Graph theory<br>• Theoretical<br>• Polynomial-time approximation algorithm | • Improve the approximation ratio of the algorithm<br>• Evaluate the impact of a different metric<br>• Include the delivery by both vehicles |
| Pugliese et al. [51] | • Graph theory | • Consider uncertainties regarding delivery resources' utilization |
| Thibbotuwawa et al. [59] | • Graph theory<br>• Decomposition method and depth-first search strategy | • Take multi-depots and battery recharging stations into account<br>• Examine options for increasing the flight's range<br>• Assess the minimization of energy consumption in the model |
| Zhu et al. [60] | • Graph theory<br>• Monte Carlo simulation<br>• Tabu search algorithm | • Calculate the drone's optimal initial freight |
| Cheng et al. [61] | • Graph theory<br>• Two nested local search algorithms<br>• Local search and outer local search | • Consider adding to the model packing time, time windows, charging the battery, and other constraints<br>• Consider the landing of drones on vehicles in motion |
| Chang and Lee [54] | • Graph theory<br>• K-means clustering technique<br>• TSP<br>• Nonlinear programming and shift-weights<br>• Simulation | • Consider constraints regarding the time window required for each delivery |
| Sacramento et al. [47] | • MIP<br>• Meta-heuristic<br>• Adaptive Large Neighborhood Search | • Include other logistics costs<br>• Consider a routing dynamic approximation caused by demand and time windows<br>• Take into account multi-drones and their interactions with trucks<br>• Examine optimal solution approaches (e.g., Dantzig–Wolfe decomposition) |
| Huang et al. [48] | • MIP<br>• ACO<br>• Neighborhood Search | • Evaluate costs' differences within VRPD and VRP for small instances<br>• Consider variability caused by demand and drone technology<br>• Consider the problem with multi-drones and multi-trucks, as well as the assignation's exchange |
| Lin et al. [49] | • MIP<br>• h-GA<br>• h-PSO | • Include time windows constraints<br>• Include uncertain conditions<br>• Evaluate other algorithms and involve a simulation approach for the synergistic dist path prob |
| Xia et al. [58] | • MIP<br>• Branch-and-price algorithm | • Consider the payload effect in the blockchain-enabled fleet-sharing platform<br>• Acknowledge demand uncertainty to include empty drones' repositioning |
| Tamke and Buscher [57] | • MILP | • Consider drone's specifications as new constraints<br>• Consider alternating the objective function to a total cost minimization to try other algorithms |
| Wang et al. [55] | • Worst case analysis<br>• Theoretical | • Evaluate other heuristics and exact algorithm approaches to solving the formulation<br>• Recognize how the algorithm performs in real-life settings |
| Luo et al. [56] | • IP<br>• Heuristic through GA | • Consider time windows' constraints<br>• Evaluate the performance of heuristics and exact algorithms within the problem |
| Ulmer and Thomas [53] | • Stochastic modeling<br>• Approximate dynamic programming | • Consider the replacement of the global parameter for state-dependent parameters |
| Choudhury et al. [62] | • Stochastic modeling<br>• Approximate dynamic programming | • Estimation of the operational costs<br>• Consider uncertainties caused by the ground vehicle network |

*3.3. Drone Delivery Scheduling Problem (DDSP)*

Scheduling problems aim to efficiently organize the delivery times for a fleet of vehicles, ensuring that commodities are delivered to their destinations on time. The problem involves assigning vehicles to predetermined routes with fixed start and finish times while minimizing costs, time, distance, or a combination of these factors. The flight range, weight capacity, and other specifications of drones vary depending on the make and model. However, generally speaking, consumer-level drones have a flight range of around 3–5 km (1.8–3.1 miles) and a weight capacity of up to 2–5 kg (4.4–11 pounds). Professional-level drones can have a longer flight range of up to 10 km (6.2 miles) or more, and a weight capacity of up to 20–30 kg (44–66 pounds) or more. However, it is important to note that drone technology is constantly evolving, and newer models with improved specifications are being introduced regularly. For drone delivery operations, scheduling decisions must be made regarding the task assignment, drone recharging, and maintenance to ensure persistent and reliable operations based on the drone's operating parameters. These problems typically consider multiple drones that can be homogeneous or heterogeneous in nature. DDSPs are generally formulated as MILP [58,63–66].

Yuan et al. [63] formulated a MILP to schedule tasks to multiple heterogeneous drones with the objective to reduce the maximum time to complete parcel delivery tasks. The problem is solved using the GA framework with three different loading methods. GA with a weight-based loading method is found to be best-suited to solve the problem, as it has a better local search performance as well as optimizes the single-flight package loading of drones while preparing for the subsequent loading. Similarly, Hazama et al. [67] and Peng et al. [68] utilized GA to solve DDSP. Hazama et al. [67] developed a model considering a drone to carry only one parcel, while Peng et al. [68] considered a drone to carry multiple parcels. Li et al. [64] developed a MILP for scheduling drone logistics but considered multi objectives: minimization of completion time while maximizing customer satisfaction. This study presented an extension of the Variable Neighborhood Search (VNS) algorithm to search for the approximate optimal solution. Lei and Chen [69] proposed an improved VNS to solve for parallel DDSP, where the procedure started with a primary solution and then continuously improved the solution using the shaking method along with an adaptive and reduced VNS.

Kim et al. [65] proposed a model that maximized the number of parcels delivered and solved the MILP model with a block-stacking-based heuristic that generated an effective solution (0% gap) very quickly for all sizes of problems. This study suggested utilizing the city building rooftops to plan an optimal operation for drone-based parcel delivery. Boysen et al. [70] identified which option is better to adopt between multiple drones or a single drone for placing on a truck. The study also investigated whether takeoff and landing stops should be identical. This study presented a MIP with the objective of minimizing the total duration of delivery tours considering an inter-modal delivery structure consisting of drones and trucks. Tavana et al. [66] modeled the DDSP as a multi-objective mixed integer mathematical programming problem and adapted the epsilon-constraint method to solve it. The model was optimized with conflicting objectives of cost and time concurrently. Torabbeigi et al. [71] proposed a two-stage stochastic scheduling approach for drones that considers drone reliability and copes with delivery failure due to mechanical or environmental reasons. The study considered two different objective functions to minimize the expected loss of the demand (ELOD) and travel time. The results demonstrated that the latter yielded a scheduling network that was around 25% more reliable. Huang et al. [72] explored the idea of scheduling a package delivery system consisting of drones and public transport via two schemes: drone-direct and drone–vehicle schemes. The study formulated a time-dependent scheduling model including the trip time, power consumption, and battery recharging.

There are several studies addressing the problem of drone charging scheduling to ensure the efficient operation of drones. Hassija et al. [73] focused on increasing the drone flight time by providing a cost-optimal drone recharging scheduling algorithm using a

game-theoretic approach. The study proposed an iterative auction-based algorithm considering both task deadline and criticality. Betti Sorbelli et al. [74] introduced the scheduling of the conflictual deliveries problem with the objective of scheduling drones while maximizing benefits subject to the battery capacity of drones. Shin et al. [75] developed an auction-based mechanism that relies on deep learning algorithms, where charging for time intervals is auctioned and assisted through a bidding process to control the charging schedule of drones. Torabbeigi et al. [76] proposed a reliable drone delivery schedule that ensures the safe drone return considering the battery charge levels while providing a minimum number of drones and delivery paths used for delivery. The study results indicate that without the battery charge rate consideration, 60% of the delivery paths become infeasible. A summary of the reviewed DDSP papers is shown in Table 3.

**Table 3.** Summary of DDSP-Related Papers.

| Author | Approach | Future Directions |
|---|---|---|
| Yuan et al. [63] | • MILP <br> • GA with weight line-based loading method | • Consider objectives other than task completion time for evaluating the scheduling algorithm |
| Hazama et al. [67] | • MILP <br> • GA considering single parcel | • Extend the problem from one drone to multiple drones |
| Peng et al. [68] | • MILP <br> • GA considering multiple parcels | • Adapt the model to similar planning and scheduling problems |
| Li et al. [64] | • MILP <br> • Extended VNS algorithm | • Incorporate more multivariate heuristic algorithms, edge computing scenarios, practical drone volume, and energy consumption models |
| Lei and Chen [69] | • MILP <br> • Adaptive reduced VNS algorithm with shaking method | • Consider environmental implications <br> • Extend the problem by adding multiple trucks and/or drones |
| Kim et al. [65] | • MILP <br> • Block-stacking-based heuristic | • Consider uncertainties, i.e., weather conditions, and battery consumption <br> • Apply other metaheuristics |
| Boysen et al. [70] | • MIP <br> • Simulated Annealing | • Solve the problem holistically to determine truck routes |
| Tavana et al. [66] | • MIP <br> • Epsilon-constraint method | • Add criteria like earliness shipping <br> • Consider multi-periods, dynamic situations, and allocation–scheduling–routing altogether <br> • Use meta-heuristic methods |
| Torabbeigi et al. [71] | • Two-stage stochastic model <br> • ELOD calculation algorithm | • Introduce uncertainty in the travel time <br> • Apply other probability distributions for drone failure function |
| Torabbeigi et al. [76] | • MILP <br> • Variable pre-possessing algorithm | • Include factors such as flight speed and environmental conditions |
| Huang et al. [72] | • Dynamic programming based <br> • Exact algorithm | • Introduce more complex public transportation network <br> • Expand the delivery area <br> • Incorporate uncertainty |
| Hassija et al. [73] | • Double Auctioning model <br> • Iterative auction-based and hash graph consensus algorithm | • Apply different algorithms to solve the problem |
| Betti Sorbelli et al. [74] | • Integer Linear Programming (ILP) model <br> • Pseudo-polynomial time optimal algorithm and approximation algorithm | • Extend the problem to multi-depot multi-truck multi-delivery scenario <br> • Incorporate late and canceled deliveries, and rescheduling deliveries during flight time |
| Shin et al. [75] | • Auction-based model <br> • Deep learning algorithm | • Formulate the problem with a multi-item auction <br> • Consider advanced auction mechanism design |

### 3.4. Drone Optimization Problem (DOP)

Optimization problem aims to find the best possible solution considering single or multiple objectives subject to certain constraints. Salama and Srinivas [77] presented a mathematical programming model that integrated the clustering and routing decisions

with the objective of minimizing the total costs and delivery time simultaneously. The study first solved the model considering a single objective; later, it considered both and obtained the best trade-off solutions. Dukkanci et al. [78] presented a nonlinear model that minimizes the overall cost and energy, both depending on speed, for drone deliveries with limited range and time-bound. Afterward, a second-order cone programming was utilized to reformulate the model, followed by the usage of perspective cuts to strengthen the model. This allowed the implementation of the off-the-shelf optimization software for solving the model.

Shavarani et al. [79] addressed both the cost and drone delivery logistics considering fuel as well as launch stations with an assumption of a uniformly distributed demand. The study proposed a mixed integer nonlinear programming (MINLP) model with the aim of minimizing the overall costs of the system. The model was solved by implementing two meta-heuristic algorithms: GA and hybrid GA, and it was found that the hybrid GA performed better than GA as a solution heuristic. Similarly, Chiang et al. [80] also used GA to solve a vehicle-drone green routing model for last-mile deliveries that investigates drone usage to save on cost and fuel consumption. The study proposed a bi-objective mixed-integer green model for minimizing the total cost and $CO_2$ emissions. The results of the research reveal that the implementation of drones for last-mile delivery is both cost-effective and environmentally friendly.

Shi et al. [81] introduced a bi-objective MIP model that optimizes both the cost and time for a multi-trip drone location routing problem, allowing pickup and delivery at the same time. The research applied a modified Non-dominated Sorting Genetic Algorithm II (NSGA-II) to solve the model, which includes double-layer coding. Khoufi et al. [82] also applied NSGA-II to solve the pickup and delivery problem for an intermittent connectivity drone network. However, it introduced an algorithm for verifying the refueling constraints that can be transformed arbitrarily without changing other parts of the algorithm. Zhang et al. [83] utilized an extended NSGA-II and adopted a local search strategy that is multi-dimensional and integrates new encoding and decoding methods as well as several operators for the crossover and mutation to optimize the economic (minimizing delivery cost and time) and environmental (minimizing energy consumption) objectives simultaneously.

Dorling et al. [84] derived a model for energy consumption and proposed a cost function that considers drone reuse for identifying sub-optimal solutions in real-life scenarios. The research developed a MILP that minimized the overall cost and delivery time and solved it by using the Simulated Annealing (SA) heuristic. The obtained results indicated that with battery weight optimization and drone reuse, over 10% of improvements could be achieved in comparison with the scenario of each drone having identical battery weight. Sajid et al. [85] proposed a joint-optimization MILP framework to solve routing and scheduling problems for drone delivery systems by implementing a combination of GA and SA with the aim of minimizing travel time. In this hybrid approach, GA utilizes stochastic crossover, as well as mutation operators to explore the search space, and at the same time, SA utilizes local search operators in this already searched space to avoid the local optima. Through result analysis, it was demonstrated that this approach outperforms all the peer approaches.

Xia et al. [58] formulated a nonlinear MIP with an objective to optimize homogeneous drone operations, which minimized the cost considering the battery wear and disposal effects. A tailored branch-and-price algorithm is implemented to solve the problem, which solved instances of up to 100 customers optimally within the allowed time limit, offering practical applications. Sawadsitang et al. [86] introduced a three-stage stochastic IP model that explicitly incorporates the uncertainty of takeoff and breakdown conditions with an aim to minimize the total delivery cost while being within the traveling distance limit. An L-shape decomposition method is adopted to handle the high complexity of the optimization problem.

A summary of the reviewed DOP papers is shown in Table 4.

**Table 4.** Summary of DOP-Related Papers.

| Author | Approach | Future Directions |
| --- | --- | --- |
| Salama and Srinivas [77] | • MILP<br>• Epsilon constraint method<br>• Iterative k-means algorithm | • Consider solving large instances using other heuristics or meta-heuristics |
| Dukkanci et al. [78] | • Second-order cone programming<br>• Exact methods | • Account truck speeds<br>• Extend the model to humanitarian applications |
| Shavarani et al. [79] | • Mixed integer non-linear programming model<br>• GA and hybrid GA | • Consider improved drone payload in capacitated models<br>• Address uncertainties by fuzzy programming approaches<br>• Assess environmental sustainability |
| Chiang et al. [80] | • MIP<br>• GA | • Consider other power sources such as fuel cells |
| Shi et al. [81] | • MIP<br>• Modified NSGA-II | • Combine underground logistics system with ground transportation |
| Khoufi et al. [82] | • MIP<br>• NSGA-II | • Optimize the refueling operations management<br>• Make the refueling time proportional to required energy of drone |
| Zhang et al. [83] | • Mixed-integer model<br>• Bi- and tri- evel heuristics | • Incorporate time windows and theory of multi-level heuristic algorithms |
| Dorling et al. [84] | • MILP<br>• Simulated annealing heuristic | • Consider the impact of weather<br>• Add time windows to locations<br>• Include maintenance cost in case of drone reuse determination |
| Xia et al. [58] | • MILP<br>• Tailored branch-and-price algorithm | • Consider the effects of different drone payloads<br>• Incorporate empty drone repositioning with fleet sharing |
| Sawadsitang et al. [86] | • Three-stage stochastic IP model<br>• Decomposition method | • Consider the uncertainty in customers' demand and traveling time<br>• Incorporate multiple-stage scenarios |

## 3.5. Urban Last Mile Problem

The ULM problem involves analyzing the final stage of drone delivery logistics in an urban environment, illustrated in Figure 6. Drones are becoming more popular in urban areas for delivery compared to traditional delivery methods due to the advantages they offer in terms of resources, such as time and money [87]. Nevertheless, there exist negative consequences of drone delivery as well, and they are a concern for the research community. With this in mind, sixteen articles were selected and classified as ULM problems. These studies addressed the positive and negative impacts of ULM and utilized a variety of techniques, including conceptual [87,88], statistical [89–92], visual, and mathematical modeling [93–102].

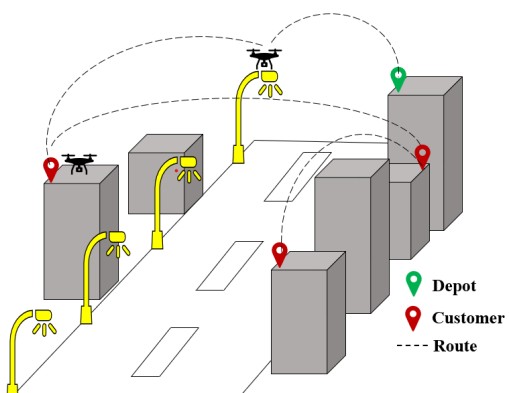

**Figure 6.** Urban Last Mile Problem—An illustration. It refers to the final leg of the delivery journey, typically from a distribution depot or hub to the end customers. Companies and transportation providers are exploring new approaches to urban delivery, such as using electric bikes, drones, and other innovative solutions, to address the urban last mile problem.

Tadic et al. [87] focused on UAV in the context of city logistics (CL). They tested and compared four CL concepts or configurations which included cooperation and flow consolidation within urban logistics. Those concepts were contrasted with one traditional concept through the generation of 10 instances, as well as the measurement of CL performances such as delivery costs, distance traveled, emissions of $CO_2$, overall delivery completion times, amount of vehicles' trips, and the loading space utilization in ground vehicles, obtaining better results within the concepts that involved a greater display of the drones, especially the last two, which were developed based on the micro-consolidation, two-echelon, and three-echelon concept. Gabani et al. [88] focused on determining where the drones' charging stations would be placed, even though a conceptual approach was also taken. They developed two frameworks; the first one consists of a truck–drone pair with the charging station for the drones placed on the upper part of the truck, while in the second one the delivery is made only by drones, and different configurations of charging stations were accommodated to fulfill the possible demand.

In relation to the statistical approach, Serrano-Hernandez et al. [89] focused on developing an Analytical Hierarchy Process (AHP) to select the transportation mode and route for a saturated city. To accomplish this, social perceptions of citizens regarding urban freight and residents' preferences were gathered and analyzed through economic, social, and environmental criteria. The results pointed out that for the residents in the city center, the drone transportation mode was the best among the rest of the alternatives (traditional van and cargo bike). Similar to the work of [89], Doole et al. [92] developed a framework to estimate the traffic density of drone-based delivery for five countries. Following that line of thought, a comparison of various forms of transportation in metropolitan areas was also made. When E-bikes (battery-assisted bicycles) were compared to drone deliveries in this instance, the cost of the E-bikes was found to be twice that of the drone deliveries. Borghetti et al. [91] concentrated on recognizing how inclined the consumers were to use the drone-based service for the last mile delivery versus other options such as a van, bicycle, and scooter using a Stated Preference (SP) Analysis with a multinomial logit model. This model was applied to a case study in a city and resulted in a high inclination to the drone's use for the customers and a profit generation after just one year of operation. Finally, Çetin et al. [90] studied society's concerns regarding the use of drones in urban areas, where problems with the environment, safety, justice, and the economy were highly mentioned. They proposed a list of mitigation measures to help overcome the concerns and improve the public's acceptance, recognizing that safety stood out as a major concern.

As mentioned before, the operations of drone delivery in an urban setting require drones to perform in highly dense areas, where it is important for society to ensure safety overall. In order to address this, studies in both visual and mathematical modeling are developed to overcome this barrier. Doole et al. [94] attempted to avoid flight conflicts and intrusions between drones while performing a delivery using time-space diagrams as a visual engine, followed by two merging strategies and algorithms that are speed-based and delay-based, both with the aim of providing a safe space for the drone to merge in the already existing traffic flow. Ren and Cheng [93] developed a model that contemplated casualties caused by low-altitude flights, as well as drone noise risks and privacy risks for third parties or a population that is not involved in the drone's flight activity. The model generated a three-dimensional grid using an image regression technique and three risk index calculations. When taking into account the flying altitude and the surrounding region, the risk index produced significant insights into the acceptability of drones as a delivery method.

Ariante et al. [96] developed a ground control system to efficiently observe and manage the UAV position for safe landing–take-off maneuvers. The implementation consists of a LiDAR system, sensor board, and wireless connectivity. Brunner et al. [99] determined the exact landing location through visual navigation. In this instance, a prototype system was developed that starts with the customer's location provided by the GPS, and then through the visual–inertial localization algorithm, the drones navigate. An extension of the work of



[99] was followed by Bahabry et al. [100]. In this case, a MIP mathematical formulation was applied with the aim to avoid drone collision and obtain optimal navigation of the multi-drone fleet, whereas the total travel time was minimized. To solve the MIP, two heuristic iterative algorithms were developed; the first one was required to define a drone's path influenced by its peer's route, whilst the second one made the path trajectories for all the drones overall. A similar problem was addressed by Mayalu et al. [101] regarding collision avoidance. Nevertheless, a different approach was used to navigate in latitudes below the urban airspace; the navigable region was decomposed into hyper-local navigation cells and used to generate 3D mappings that enhanced the trajectory planning. In the case of Li et al. [102], the flight scheduling and conflict resolution were discussed through a deterministic clustering-based single path planning, which results in four traffic management models. Each model represents a different aggregate configuration regarding the flight priority and possible path defined for the drone. For the most complex, an MILP is used to formulate the overall problem with the objective of minimizing the system's total cost conformed by the mission's delay cost and path cost.

Zang et al. [97] presented a ULM delivery problem with drones and trucks where three mathematical models—mixed integer, bi-level, and three-level programming—were formulated. The study developed two heuristics: bi- and tri-level heuristics that take significantly less computation time in comparison with CPLEX to solve the problems.Resat [98] proposed a hybrid solution methodology including a multi-criteria decision-making (MCDM) system and a MILP optimization model for ensuring a sustainable last-mile delivery in urban areas. The MCDM system identified the best logistics providers based on various performance parameters, and the MILP minimized transportation costs and total carbon dioxide emissions by using the epsilon constraint method. An interesting approach was proposed by Kuru [103] on the matter of fully automated UAVs (FAUAVs) in urban areas, especially within the concept of smart cities (SCs). They developed a framework that follows the requirements and controls in real-time, along with all the resource limitations regarding the FAUAVs in a mission. The framework was based on a decentralized agent-based control architecture to integrate both concepts, FAUVs, and SCs. To fulfill this, the FAUVs are conformed of cameras, LIDAR, radar, sensors, inertial measurement units, inertial navigation systems, and communication systems that are able to interact with the SCs' technology.

Table 5 provides an overview of the reviewed papers under the context of ULM.

**Table 5.** Summary of ULM Papers.

| Author | Approach | Future Directions |
|---|---|---|
| Tadic et al. [87] | • CL conceptual models<br>• Performance evaluation through test instance generation | • Develop a financial risk assessment for the parts involved<br>• Consider the CL concepts in a dynamic and stochastic environment<br>• Consider other CL concepts and combinations<br>• Generate new models oriented to the implementation of more complex CL concepts with drones |
| Gabani et al. [88] | • Conceptual models<br>• Case scenarios | • Consider stakeholders' impact on the framework implementation<br>• Expand the framework to the computational simulation<br>• Consider implementation feasibility and profitability |
| Serrano-Hernandez et al. [89] | • Statistical<br>• Survey and multi-criteria analysis (AHP) | • Include more stakeholders in the analysis (carriers, owners, and local authorities) |
| Doole et al. [92] | • Statistical<br>• Estimation framework and forecasting<br>• Case study | • Expand financial assessment, including factors such as drone's landing area and charging station |

**Table 5.** *Cont.*

| Author | Approach | Future Directions |
|---|---|---|
| Borghetti et al. [91] | • Statistical<br>• Stated Preference (SP) survey<br>• Multinomial logit model Numerical case study, financial feasibility analysis | • Consider legal regulations and limitations in the drone's route planning<br>• Explore landing strategies for dense urban areas<br>• Include battery performance and its impact in the drone's performance<br>• Explore end-user recognition<br>• Use of multi-criteria analysis to recognize the environmental impact of drone's use |
| Çetin et al. [90] | • Statistical<br>• Survey, brainstorming, and safety operational risk assessment | • Implement the proposed mitigations<br>• Expand the techniques to measure and recollect data for the list's improvement |
| Doole et al. [94] | • Time–space diagram<br>• Speed-based and delay-based algorithms Simulation | • Include other real factors such as meteorological events<br>• Consider drone's flight information to avoid false recognition<br>• Enhance the street network to be non-orthogonal |
| Ren and Cheng [93] | • Pixel regression mode<br>• Risk assessment index model | • Explore verification techniques for real-life applications of the model<br>• Include other drones' internal (endurance) and external factors (weather and airspace) |
| Ariante et al. [96] | • 2D LiDAR-based Ground System | • Enhance the mechanical structure<br>• Improve the calibration strategy |
| Zang et al. [97] | • MIP<br>• Bi-level and three-level programming | • Include time window<br>• Consider multi-level heuristic algorithm to solve the problem |
| Resat [98] | • MILP<br>• MCDM | • Include drone's characteristics as models' constraints<br>• Consider sensitivity analysis to recognize the parameters that influence the sustainability scores |
| Bahabry et al. [100] | • MILP<br>• Two heuristics algorithms | • Consider research approaches that can enhance solutions for real-time cases (e.g., artificial intelligence)<br>• Consider the inclusion of a ground transit network to help the drone's energy endurance |
| Mayalu et al. [101]<br><br>Li et al. [102] | • 3D-Mapping<br>• 3D-Tiles navigation format<br>• Robot Operating System (ROS)<br>• Deterministic clustering-based path planning<br>• Saturated Fast-Marching Square (Saturated FM2) algorithm<br>• MILP, linearization<br>• Batch optimization algorithm | • Expand trajectory planning for drone traffic management applications<br><br>• Expand the use of other heuristics approaches to solve the model formulation<br>• Incorporate stochastic factors<br>• Recognize the effect of population density, building concentration, and terrain types on the model's performance |
| Brunner et al. [99] | • GPS-based navigation<br>• ROS, PX4, Ardupilot<br>• Vision-based localization algorithms and simulation | • Include collision avoidance in the model with data retrieval<br>• Add building scan detection to detect the landing field<br>• Expand the study to incorporate package handling |
| Kuru [103] | • Decentralized agent-based control architecture<br>• Simulation | • Develop regulations to frame the FAUAVs operations in SCs<br>• Improve the communication technology with the FAUAVs and SCs<br>• Consider other options for interference management and jamming avoidance techniques<br>• Include sky pollution reduction within the UAV route planning |

## 4. Challenges

### 4.1. Technological

The battery and flight-time duration limit the range of drones. As explained by Kirschstein [104], one challenging problem is reducing the power usage and minimizing the impact of wind direction, trip speed, and customer density, which ends up affecting the efficiency of drone-delivery systems. Moreover, how to correctly aim at the greater density locations needs to be further discussed, as it may offer cost-saving potential in comparison to drone implementations for rural settings. In this regard, the number of customers, traffic circumstances, and even battery deterioration have to be addressed. Another issue connected to battery duration is determining the right size of the required facilities, which is addressed by Aiello et al. [105]. Further, Torabbeigi et al. [76] mentioned the need for tackling the impact of the payload on the battery consumption rate (BCR) and flying time. Multiple charging stations will expand the last mileage delivery coverage, which can be a factor to consider for a further impact analysis [59]. Moreover, the location of the charging station must be taken into account in the analysis [35]. Hong et al. [106] identified the need to construct a location model for a spatially dispersed charging station. Additionally, the drone charging time is a factor to consider since it will influence the time constraint in the mathematical formulation. In this case, Daknama and Kraus [50] suggested that factors, e.g., partial charge and charging speed, should be considered for the drone charging process in order to further adjust it to real scenarios.

### 4.2. Social Perception

According to a Poland-based study, 43% of the population was skeptical about the implementation of drones, which implies the existence of social barriers to the adoption of drone parcel delivery services [107]. Another study conducted in the urban areas of Australia found that traditional postal services are preferred to drone deliveries despite the recent advancement in e-commerce and technology [17]. Social anxiety about automation contributes to the skepticism about drones [108]. People are concerned that the usage of drones will make traditional retail disappear to a great extent, which will eventually lead to job losses, increased stress levels, decreased social interactions, and the creation of an elite mobility regime [21]. Social equity is considered to be another significant barrier to the implementation of drone delivery. It is feared that drones will only be limited to rich households, as there remains uncertainty regarding their affordability [109]. Public knowledge about drone technology and operations can help consumers better understand the possible benefits of adopting drones as a delivery mode. Mass media channels can be used as a tool to educate the public about drones, considering that media have a positive effect on the public attitude [110].

### 4.3. Privacy and Safety

The biggest challenges of drone implementation for delivery are the possible privacy violations, safety, and ethical issues. Drones are exposed to various security attacks while communicating with ground facilities via open channels [111]. Delivery drones are susceptible to privacy infringement, as they have information about consumers and are also equipped with cameras. Drones might be hacked for stealing personal information and scamming people [112]. Another concern is safety, as drones carrying a parcel can fall into unintended destinations, causing severe harm to people. The primary causes for ground impact drone accidents are a shortage of battery power and partial failure of the rotor or battery [113]. Drones can collide with each other for a loss of communication or power and hardware or software malfunctions [114]. Moreover, there is a potential threat of drones being weaponized for terrorism or smuggling [21]. A possible intrusion into privacy and safety risks negatively impacts peoples' attitudes toward drone-aided delivery. This situation takes a higher relevance in the case of the marginalized population [115]. To tackle this, advanced technologies utilized in drone design should be highlighted to minimize their anxiety regarding security risks [110]. Privacy and safety challenges can be addressed

by establishing no-fly zones for drones, using advanced encryption to restrict cyber-attacks, limiting camera usage, implementing a no-access rule to any recordings during a drone flight, limiting minimum altitudes, and hovering drone flights [90]. The zoning approach can be implemented as a systematic solution for drone deployment on a large scale [116].

*4.4. Environmental Concerns*

Environmental concerns affect the user's attitude toward drone delivery. Hence, it must be ensured that environmentally conscious best practices are adopted for drone operations [117]. It is found that employing drones on a large scale is likely to reduce pollution more effectively in rural areas than urban areas [118]. Drones also cause noise pollution, $CO_2$ emission and visual pollution [103,119–121]. A study evaluated soundscapes of different locations, considering the effect of drone noise on traffic. In areas that were adjacent to busy roads, the noise generated by drones was masked by the traffic noise, causing a sound annoyance only 1.13 times more than without any drone noise, in comparison to other areas with less busy roads where the sound annoyance created by drones was 6.4 times higher. This indicates that planning drone operation routes near busy roads can significantly diminish the noise pollution it causes [122]. Concerning visual pollution, a challenge for operators is improving the drone's route planning [103]. Adverse weather conditions (i.e., windstorms, snowstorms, poor visibility, and thunderstorm) pose a big challenge to smooth drone delivery [108,117]. Furthermore, drones can collide with birds and harm other animals [90,109]. To mitigate these challenges, the adoption of renewable energy sources for charging drone batteries and designing eco-friendly or hybrid drones by implementing recycling needs to be encouraged [90].

# 5. Concluding Discussions and Limitations

## 5.1. Concluding Discussions

This study reviewed the main contributions in the literature related to drone-aided delivery between 2015 and 2022. The volume of literature published during this period confirms this topic's potential and researchers' interest in drone deliveries. We classified the literature into four sections and analyzed each section based on methodologies, objectives, and future research potential. We also summarized all the literature of a section in a table, which is helpful for quickly comparing different studies and detecting the unexplored areas of each of them.

Future studies should consider the remaining battery capacity after a flight, speed variations, and battery swapping schedules to make the models more relevant to real-world scenarios. As the speed of drones increases, flight time, battery life, and cost will be impacted, and these factors need to be studied. To make the models more adaptable to possible technological advancements, some constraints could be relaxed. For instance, the drone flight range or capacity constraints could be relaxed to observe how they affect the optimal solutions of a model.

Since drone operations are susceptible to uncertainties such as weather variability and man-made disasters, the models should incorporate them. Researchers should address questions such as what a drone should do if it encounters such uncertainties during the delivery flight. Should it return to the base or the nearest charging station, or identify the nearest safe place to land? By answering such questions, the models can be improved to better reflect real-world scenarios.

The reviewed literature exhibited an interesting trend, as a majority of studies relied on randomly generated data to validate their formulated models and techniques. Such data was often collected from small-scale drone tests, which limits its reliability in predicting the performance of large-scale operations. Consequently, managerial conclusions drawn from these studies may not hold significant value for real-life drone operations on a larger scale. To ensure robustness, the studies should perform a sensitivity analysis using a wide range of values. Furthermore, most studies assumed a small fleet of drones and a limited number of customers, highlighting the need for research that addresses larger-scale operations.

As of now, the operational capabilities of drones are restricted by their capacity, battery life, and sensitivity to weather conditions, as well as the maximum payload they can carry. However, with ongoing technological advancements, it is reasonable to anticipate that drones will soon overcome these limitations and be readily available for commercial long-range operations. This will necessitate having drones that are durable enough to fly for extended periods of time and serve a larger customer base, which in turn will heighten the need for more frequent maintenance and increased safety standards. In the future, safety can be integrated into the modeling of drone operations. The evolution of drone technology may also make current challenges outdated, and with commercialization, profitability is likely to become an essential factor to consider.

As drone delivery becomes more widespread, it is important to consider customer preferences, satisfaction, and dissatisfaction in the existing system. Researchers should explore how to incorporate these factors into operational models. For example, what if a customer prefers to be present during the delivery, has a preferred time of delivery, or fails to show up for the delivery? Additionally, if a high demand prevents some delivery orders from being fulfilled, researchers should examine ways to reflect this in the models.

Further questions that should be explored include how to determine the order of serving customers when multiple requests need to be fulfilled by one drone, whether drones should wait until the next request if there are no immediate customers to serve, and how many drones should be used to meet the demands. Overall, incorporating customer preferences and satisfaction into drone delivery models will be critical for ensuring the success of this emerging technology.

The challenges of drone delivery in urban environments have been identified and classified into four parts, with possible ways to overcome them discussed. This will enable practitioners to address the challenges and overcome barriers to implementing drone-aided deliveries. As drone delivery becomes more prevalent in urban areas, airspace will become more congested, increasing the risk of collisions. Incorporating this factor into the model and objective functions could be useful. Another significant challenge in drone operation that needs to be modeled is noise. The navigation system of drones may encounter unknown barriers in urban areas. To address this challenge and improve drone recognition systems, new approaches in artificial intelligence need to be introduced.

### 5.2. Limitations

This study has certain limitations. The literature review may not have been exhaustive, as the search was conducted using specific keywords and databases. However, we intentionally curated recent literature on drone-aided delivery under the context of urban logistics. Future studies could benefit from expanding their search by using broader databases and incorporating additional relevant keywords. Additionally, future research on drone-aided delivery should consider integrating drones with existing urban transportation systems and addressing the potential challenges to increase the acceptance and implementation of drone delivery.

**Author Contributions:** Conceptualization, X.L.; methodology, A.S. and J.T.; writing—original draft preparation, A.S., J.T., and M.M.F.; writing—review and editing, Li, X, A.S., J.T., and M.M.F.; supervision and funding acquisition, X.L. All authors have read and agreed to the published version of the manuscript.

**Funding:** This research received no external funding.

**Data Availability Statement:** No new data were created or analyzed in this study. Data sharing is not applicable to this article.

**Conflicts of Interest:** he authors declare no conflict of interest.

1.　Thibbotuwawa, A.; Bocewicz, G.; Nielsen, P.; Banaszak, Z. Unmanned aerial vehicle routing problems: A literature review. *Appl. Sci.* **2020**, *10*, 4504. [CrossRef]

2.　Rejeb, A.; Rejeb, K.; Simske, S.; Treiblmaier, H. Humanitarian drones: A review and research agenda. *Internet Things* **2021**, *16*, 100434. [CrossRef]

3.　Lin, I.C.; Lin, T.H.; Chang, S.H. A decision system for routing problems and rescheduling issues using Unmanned Aerial Vehicles. *Appl. Sci.* **2022**, *12*, 6140. [CrossRef]

4.　Almuhaideb, S.; Alhussan, T.; Alamri, S.; Altwaijry, Y.; Aljarbou, L.; Alrayes, H. Optimization of truck-drone parcel delivery using metaheuristics. *Appl. Sci.* **2021**, *11*, 6443. [CrossRef]

5.　Mohsan, S.A.H.; Zahra, Q.u.A.; Khan, M.A.; Alsharif, M.H.; Elhaty, I.A.; Jahid, A. Role of drone technology helping in alleviating the COVID-19 pandemic. *Micromachines* **2022**, *13*, 1593. [CrossRef]

6.　Szász, L.; Bálint, C.; Csíki, O.; Nagy, B.Z.; Rácz, B.G.; Csala, D.; Harris, L.C. The impact of COVID-19 on the evolution of online retail: The pandemic as a window of opportunity. *J. Retail. Consum. Serv.* **2022**, *69*, 103089. [CrossRef]

7.　Bilinska-Reformat, K.; Dewalska-Opitek, A. E-commerce as the predominant business model of fast fashion retailers in the era of global COVID 19 pandemics. *Procedia Comput. Sci.* **2021**, *192*, 2479–2490. [CrossRef]

8.　Raj, A.; Mukherjee, A.A.; de Sousa Jabbour, A.B.L.; Srivastava, S.K. Supply chain management during and post-COVID-19 pandemic: Mitigation strategies and practical lessons learned. *J. Bus. Res.* **2022**, *142*, 1125–1139. [CrossRef]

9.　Gupta, A.; Afrin, T.; Scully, E.; Yodo, N. Advances of UAVs toward future transportation: The state-of-the-art, challenges, and opportunities. *Future Transp.* **2021**, *1*, 326–350. [CrossRef]

10.　Suguna, M.; Shah, B.; Raj, S.K.; Suresh, M. A study on the influential factors of the last mile delivery projects during COVID-19 era. *Oper. Manag. Res.* **2021**, *15*, 399–412. [CrossRef]

11.　Sorooshian, S.; Sharifabad, S.K.; Parsaee, M.; Afshari, A.R. Toward a modern last-mile delivery: Consequences and obstacles of intelligent technology. *Appl. Syst. Innov.* **2022**, *5*, 82. [CrossRef]

12.　Merkert, R.; Bliemer, M.C.; Fayyaz, M. Consumer preferences for innovative and traditional last-mile parcel delivery. *Int. J. Phys. Distrib. Logist. Manag.* **2022**, *52*, 261–284. [CrossRef]

13.　Mohsan, S.A.H.; Khan, M.A.; Noor, F.; Ullah, I.; Alsharif, M.H. Towards the unmanned aerial vehicles (UAVs): A comprehensive review. *Drones* **2022**, *6*, 147. [CrossRef]

14.　Ayamga, M.; Akaba, S.; Nyaaba, A.A. Multifaceted applicability of drones: A review. *Technol. Forecast. Soc. Chang.* **2021**, *167*, 120677. [CrossRef]

15.　Hassanalian, M.; Abdelkefi, A. Classifications, applications, and design challenges of drones: A review. *Prog. Aerosp. Sci.* **2017**, *91*, 99–131. [CrossRef]

16.　Beloev, I.H. A review on current and emerging application possibilities for unmanned aerial vehicles. *Acta Technol. Agric.* **2016**, *19*, 70–76. [CrossRef]

17.　Merkert, R.; Bushell, J. Managing the drone revolution: A systematic literature review into the current use of airborne drones and future strategic directions for their effective control. *J. Air Transp. Manag.* **2020**, *89*, 101929. [CrossRef]

18.　Benarbia, T.; Kyamakya, K. A literature review of drone-based package delivery logistics systems and their implementation feasibility. *Sustainability* **2022**, *14*, 360. [CrossRef]

19.　Büyüközkan, G.; Ilıcak, Ö. Smart urban logistics: Literature review and future directions. *Socio-Econ. Plan. Sci.* **2022**, *81*, 101197. [CrossRef]

20.　Mohd Noor, N.; Abdullah, A.; Hashim, M. Remote sensing UAV/drones and its applications for urban areas: A review. *IOP Conf. Ser. Earth Environ. Sci.* **2018**, *169*, 012003. [CrossRef]

21.　Kellermann, R.; Biehle, T.; Fischer, L. Drones for parcel and passenger transportation: A literature review. *Transp. Res. Interdiscip. Perspect.* **2020**, *4*, 100088. [CrossRef]

22.　Khoufi, I.; Laouiti, A.; Adjih, C. A survey of recent extended variants of the traveling salesman and vehicle routing problems for unmanned aerial vehicles. *Drones* **2019**, *3*, 66. [CrossRef]

23.　Chung, S.H.; Sah, B.; Lee, J. Optimization for drone and drone-truck combined operations: A review of the state of the art and future directions. *Comput. Oper. Res.* **2020**, *123*, 105004. [CrossRef]

24.　Rojas Viloria, D.; Solano-Charris, E.L.; Muñoz-Villamizar, A.; Montoya-Torres, J.R. Unmanned aerial vehicles/drones in vehicle routing problems: A literature review. *Int. Trans. Oper. Res.* **2021**, *28*, 1626–1657. [CrossRef]

25.　Moshref-Javadi, M.; Winkenbach, M. Applications and research avenues for drone-based models in logistics: A classification and review. *Expert Syst. Appl.* **2021**, *177*, 114854. [CrossRef]

26.　Rejeb, A.; Rejeb, K.; Simske, S.J.; Treiblmaier, H. Drones for supply chain management and logistics: A review and research agenda. *Int. J. Logist. Res. Appl.* **2021**, 1–24. [CrossRef]

27.　Macrina, G.; Di Puglia Pugliese, L.; Guerriero, F.; Laporte, G. Drone-aided routing: A literature review. *Transp. Res. Part C Emerg. Technol.* **2020**, *120*, 102762. [CrossRef]

28.　Spasojevic, B.; Lohmann, G.; Scott, N. Air transport and tourism—A systematic literature review (2000–2014). *Curr. Issues Tour.* **2018**, *21*, 975–997. [CrossRef]

29.　Tranfield, D.; Denyer, D.; Smart, P. Towards a methodology for developing evidence-informed management knowledge by means of systematic review. *Br. J. Manag.* **2003**, *14*, 207–222. [CrossRef]

30. Jacso, P. The h-index, h-core citation rate and the bibliometric profile of the Scopus database. *Online Inf. Rev.* **2011**, *35*, 492–501. [CrossRef]
31. Yurek, E.E.; Ozmutlu, H.C. A decomposition-based iterative optimization algorithm for traveling salesman problem with drone. *Transp. Res. Part C Emerg. Technol.* **2018**, *91*, 249–262. [CrossRef]
32. Murray, C.C.; Chu, A.G. The flying sidekick traveling salesman problem: Optimization of drone-assisted parcel delivery. *Transp. Res. Part C Emerg. Technol.* **2015**, *54*, 86–109. [CrossRef]
33. Cavani, S.; Iori, M.; Roberti, R. Exact methods for the traveling salesman problem with multiple drones. *Transp. Res. Part C Emerg. Technol.* **2021**, *130*, 103280. [CrossRef]
34. Boccia, M.; Masone, A.; Sforza, A.; Sterle, C. An exact approach for a variant of the FS-TSP. *Transp. Res. Procedia* **2021**, *52*, 51–58. [CrossRef]
35. Kim, S.; Moon, I. Traveling salesman problem with a drone station. *IEEE Trans. Syst. Man Cybern. Syst.* **2019**, *49*, 42–52. [CrossRef]
36. Bouman, P.; Agatz, N.; Schmidt, M. Dynamic programming approaches for the traveling salesman problem with drone. *Networks* **2018**, *72*, 528–542. [CrossRef]
37. Ha, Q.M.; Deville, Y.; Pham, Q.D.; Hà, M.H. On the min-cost traveling salesman problem with drone. *Transp. Res. Part C Emerg. Technol.* **2018**, *86*, 597–621. [CrossRef]
38. Marinelli, M.; Caggiani, L.; Ottomanelli, M.; Dell'Orco, M. *En route* truck-drone parcel delivery for optimal vehicle routing strategies. *IET Intell. Transp. Syst.* **2018**, *12*, 253–261. [CrossRef]
39. de Freitas, J.C.; Penna, P.H.V. A variable neighborhood search for flying sidekick traveling salesman problem. *Int. Trans. Oper. Res.* **2019**, *27*, 267–290. [CrossRef]
40. Ha, Q.M.; Deville, Y.; Pham, Q.D.; Hà, M.H. A hybrid genetic algorithm for the traveling salesman problem with drone. *J. Heuristics* **2020**, *26*, 219–247. [CrossRef]
41. Kitjacharoenchai, P.; Ventresca, M.; Moshref-Javadi, M.; Lee, S.; Tanchoco, J.M.; Brunese, P.A. Multiple traveling salesman problem with drones: Mathematical model and heuristic approach. *Comput. Ind. Eng.* **2019**, *129*, 14–30. [CrossRef]
42. Raj, R.; Murray, C. The multiple flying sidekicks traveling salesman problem with variable drone speeds. *Transp. Res. Part C Emerg. Technol.* **2020**, *120*, 102813. [CrossRef]
43. Baniasadi, P.; Foumani, M.; Smith-Miles, K.; Ejov, V. A transformation technique for the clustered generalized traveling salesman problem with applications to logistics. *Eur. J. Oper. Res.* **2020**, *285*, 444–457. [CrossRef]
44. Dell'Amico, M.; Montemanni, R.; Novellani, S. Matheuristic algorithms for the parallel drone scheduling traveling salesman problem. *Ann. Oper. Res.* **2020**, *289*, 211–226. [CrossRef]
45. Mathew, N.; Smith, S.L.; Waslander, S.L. Planning paths for package delivery in heterogeneous multirobot teams. *IEEE Trans. Autom. Sci. Eng.* **2015**, *12*, 1298–1308. [CrossRef]
46. Saleu, R.G.M.; Deroussi, L.; Feillet, D.; Grangeon, N.; Quilliot, A. The parallel drone scheduling problem with multiple drones and vehicles. *Eur. J. Oper. Res.* **2022**, *300*, 571–589. [CrossRef]
47. Sacramento, D.; Pisinger, D.; Ropke, S. An adaptive large neighborhood search metaheuristic for the vehicle routing problem with drones. *Transp. Res. Part C Emerg. Technol.* **2019**, *102*, 289–315. [CrossRef]
48. Huang, S.H.; Huang, Y.H.; Blazquez, C.A.; Chen, C.Y. Solving the vehicle routing problem with drone for delivery services using an ant colony optimization algorithm. *Adv. Eng. Inform.* **2022**, *51*, 101536. [CrossRef]
49. Lin, M.; Lyu, J.Y.; Gao, J.J.; Li, L.Y. Model and hybrid algorithm of collaborative distribution system with multiple drones and a truck. *Sci. Program.* **2020**, *2020*, 8887057. [CrossRef]
50. Daknama, R.; Kraus, E. Vehicle Routing with Drones. *arXiv* **2017**, arXiv:1705.06431.
51. Pugliese, L.D.P.; Guerriero, F.; Macrina, G. Using drones for parcels delivery process. *Procedia Manuf.* **2020**, *42*, 488–497. [CrossRef]
52. Othman, M.S.B.; Shurbevski, A.; Karuno, Y.; Nagamochi, H. Routing of carrier-vehicle systems with dedicated last-stretch delivery vehicle and fixed carrier route. *J. Inf. Process.* **2017**, *25*, 655–666. [CrossRef]
53. Ulmer, M.W.; Thomas, B.W. Same-day delivery with heterogeneous fleets of drones and vehicles. *Networks* **2018**, *72*, 475–505. [CrossRef]
54. Chang, Y.S.; Lee, H.J. Optimal delivery routing with wider drone-delivery areas along a shorter truck-route. *Expert Syst. Appl.* **2018**, *104*, 307–317. [CrossRef]
55. Wang, X.; Poikonen, S.; Golden, B. The vehicle routing problem with drones: Several worst-case results. *Optim. Lett.* **2017**, *11*, 679–697. [CrossRef]
56. Luo, Z.; Liu, Z.; Shi, J. A two-echelon cooperated routing problem for a ground vehicle and its carried unmanned aerial vehicle. *Sensors* **2017**, *17*, 1144. [CrossRef]
57. Tamke, F.; Buscher, U. A branch-and-cut algorithm for the vehicle routing problem with drones. *Transp. Res. Part B Methodol.* **2021**, *144*, 174–203. [CrossRef]
58. Xia, Y.; Zeng, W.; Xing, X.; Zhan, Y.; Tan, K.H.; Kumar, A. Joint optimisation of drone routing and battery wear for sustainable supply chain development: A mixed-integer programming model based on blockchain-enabled fleet sharing. *Ann. Oper. Res.* **2021**, 1–39. [CrossRef]
59. Thibbotuwawa, A.; Bocewicz, G.; Nielsen, P.; Zbigniew, B. Planning deliveries with UAV routing under weather forecast and energy consumption constraints. *IFAC-PapersOnLine* **2019**, *52*, 820–825. [CrossRef]

60. Zhu, X.; Yan, R.; Peng, R.; Zhang, Z. Optimal routing, loading and aborting of UAVs executing both visiting tasks and transportation tasks. *Reliab. Eng. Syst. Saf.* **2020**, *204*, 107132. [CrossRef]

61. Cheng, C.; Adulyasak, Y.; Rousseau, L.M. Drone routing with energy function: Formulation and exact algorithm. *Transp. Res. Part B Methodol.* **2020**, *139*, 364–387. [CrossRef]

62. Choudhury, S.; Solovey, K.; Kochenderfer, M.J.; Pavone, M. Efficient large-scale multi-drone delivery using transit networks. *J. Artif. Intell. Res.* **2021**, *70*, 757–788. [CrossRef]

63. Yuan, X.; Zhu, J.; Li, Y.; Huang, H.; Wu, M. An enhanced genetic algorithm for unmanned aerial vehicle logistics scheduling. *IET Commun.* **2021**, *15*, 1402–1411. [CrossRef]

64. Li, Y.; Yuan, X.; Zhu, J.; Huang, H.; Wu, M. Multi-objective scheduling of logistics UAVs based on simulated annealing. *Commun. Comput. Inf. Sci.* **2020**, *1163*, 287–298. [CrossRef]

65. Kim, J.; Moon, H.; Jung, H. Drone-based parcel delivery using the rooftops of city buildings: Model and solution. *Appl. Sci.* **2020**, *10*, 4362. [CrossRef]

66. Tavana, M.; Khalili-Damghani, K.; Santos-Arteaga, F.J.; Zandi, M.H. Drone shipping versus truck delivery in a cross-docking system with multiple fleets and products. *Expert Syst. Appl.* **2017**, *72*, 93–107. [CrossRef]

67. Hazama, Y.; Iima, H.; Karuno, Y.; Mishima, K. Genetic algorithm for scheduling of parcel delivery by drones. *J. Adv. Mech. Des. Syst. Manuf.* **2021**, *15*, 1–12. [CrossRef]

68. Peng, K.; Du, J.; Lu, F.; Sun, Q.; Dong, Y.; Zhou, P.; Hu, M. A hybrid genetic algorithm on routing and scheduling for vehicle-assisted multi-drone parcel delivery. *IEEE Access* **2019**, *7*, 49191–49200. [CrossRef]

69. Lei, D.; Chen, X. An improved variable neighborhood search for parallel drone scheduling traveling salesman problem. *Appl. Soft Comput.* **2022**, *127*, 109416. [CrossRef]

70. Boysen, N.; Briskorn, D.; Fedtke, S.; Schwerdfeger, S. Drone delivery from trucks: Drone scheduling for given truck routes. *Networks* **2018**, *72*, 506–527. [CrossRef]

71. Torabbeigi, M.; Lim, G.J.; Kim, S.J. Drone delivery schedule optimization considering the reliability of drones. In Proceedings of the 2018 International Conference on Unmanned Aircraft Systems, ICUAS 2018, Dallas, TX, USA, 12–15 June 2018; pp. 1048–1053. [CrossRef]

72. Huang, H.; Savkin, A.V.; Huang, C. Scheduling of a parcel delivery system consisting of an aerial drone interacting with public transportation vehicles. *Sensors* **2020**, *20*, 2045. [CrossRef] [PubMed]

73. Hassija, V.; Saxena, V.; Chamola, V. Scheduling drone charging for multi-drone network based on consensus time-stamp and game theory. *Comput. Commun.* **2020**, *149*, 51–61. [CrossRef]

74. Betti Sorbelli, F.; Corò, F.; Das, S.K.; Palazzetti, L.; Pinotti, C.M. On the scheduling of conflictual deliveries in a last-mile delivery scenario with truck-carried drones. *Pervasive Mob. Comput.* **2022**, *87*, 101700. [CrossRef]

75. Shin, M.; Kim, J.; Levorato, M. Auction-based charging scheduling with deep learning framework for multi-drone networks. *IEEE Trans. Veh. Technol.* **2019**, *68*, 4235–4248. [CrossRef]

76. Torabbeigi, M.; Lim, G.J.; Kim, S.J. Drone delivery scheduling optimization considering payload-induced battery consumption rates. *J. Intell. Robot. Syst.* **2020**, *97*, 471–487. [CrossRef]

77. Salama, M.; Srinivas, S. Joint optimization of customer location clustering and drone-based routing for last-mile deliveries. *Transp. Res. Part C Emerg. Technol.* **2020**, *114*, 620–642. [CrossRef]

78. Dukkanci, O.; Kara, B.Y.; Bektaş, T. Minimizing energy and cost in range-limited drone deliveries with speed optimization. *Transp. Res. Part C Emerg. Technol.* **2021**, *125*, 102985. [CrossRef]

79. Shavarani, S.M.; Nejad, M.G.; Rismanchian, F.; Izbirak, G. Application of hierarchical facility location problem for optimization of a drone delivery system: A case study of Amazon prime air in the city of San Francisco. *Int. J. Adv. Manuf. Technol.* **2018**, *95*, 3141–3153. [CrossRef]

80. Chiang, W.C.; Li, Y.; Shang, J.; Urban, T.L. Impact of drone delivery on sustainability and cost: Realizing the UAV potential through vehicle routing optimization. *Appl. Energy* **2019**, *242*, 1164–1175. [CrossRef]

81. Shi, Y.; Lin, Y.; Li, B.; Li, R.Y.M. A bi-objective optimization model for the medical supplies' simultaneous pickup and delivery with drones. *Comput. Ind. Eng.* **2022**, *171*, 108389. [CrossRef]

82. Khoufi, I.; Laouiti, A.; Adjih, C.; Hadded, M. UAV trajectory optimization for data pick up and delivery with time windows. *Drones* **2021**, *5*, 27. [CrossRef]

83. Zhang, S.; Liu, S.; Xu, W.; Wang, W. A novel multi-objective optimization model for the vehicle routing problem with drone delivery and dynamic flight endurance. *Comput. Ind. Eng.* **2022**, *173*, 108679. [CrossRef]

84. Dorling, K.; Heinrichs, J.; Messier, G.G.; Magierowski, S. Vehicle routing problems for drone delivery. *IEEE Trans. Syst. Man Cybern. Syst.* **2017**, *47*, 70–85. [CrossRef]

85. Sajid, M.; Mittal, H.; Pare, S.; Prasad, M. Routing and scheduling optimization for UAV assisted delivery system: A hybrid approach. *Appl. Soft Comput.* **2022**, *126*, 109225. [CrossRef]

86. Sawadsitang, S.; Niyato, D.; Tan, P.S.; Wang, P. Joint ground and aerial package delivery services: A stochastic optimization approach. *IEEE Trans. Intell. Transp. Syst.* **2019**, *20*, 2241–2254. [CrossRef]

87. Tadić, S.; Kovač, M.; Čokorilo, O. The application of drones in city logistics concepts. *Promet-Traffic Transp.* **2021**, *33*, 451–462. [CrossRef]

88. Gabani, P.R.; Gala, U.B.; Narwane, V.S.; Raut, R.D.; Govindarajan, U.H.; Narkhede, B.E. A viability study using conceptual models for last mile drone logistics operations in populated urban cities of India. *IET Collab. Intell. Manuf.* **2021**, *3*, 262–272. [CrossRef]

89. Serrano-Hernandez, A.; Ballano, A.; Faulin, J. Selecting freight transportation modes in last-mile urban distribution in pamplona (Spain): An option for drone delivery in smart cities. *Energies* **2021**, *14*, 4748. [CrossRef]

90. Çetin, E.; Cano, A.; Deransy, R.; Tres, S.; Barrado, C. Implementing mitigations for improving societal acceptance of urban air mobility. *Drones* **2022**, *6*, 28. [CrossRef]

91. Borghetti, F.; Caballini, C.; Carboni, A.; Grossato, G.; Maja, R.; Barabino, B. The use of drones for last-Mile delivery: A numerical case study in Milan, Italy. *Sustainability* **2022**, *14*, 1766. [CrossRef]

92. Doole, M.; Ellerbroek, J.; Hoekstra, J. Estimation of traffic density from drone-based delivery in very low level urban airspace. *J. Air Transp. Manag.* **2020**, *88*, 101862. [CrossRef]

93. Ren, X.; Cheng, C. Model of third-party risk index for unmanned aerial vehicle delivery in urban environment. *Sustainability* **2020**, *12*, 8318. [CrossRef]

94. Doole, M.; Ellerbroek, J.; Hoekstra, J.M. Investigation of merge assist policies to improve safety of drone traffic in a constrained urban airspace. *Aerospace* **2022**, *9*, 120. [CrossRef]

95. Çetin Kaya, Y.; Kaya, M.; Akdağ, A. Determining optimal route for medication delivery of COVID-19 patients with drone. *Gazi Univ. J. Sci. Part C Des. Technol.* **2021**, *9*, 478–491. [CrossRef]

96. Ariante, G.; Ponte, S.; Papa, U.; Greco, A.; Core, G.D. Ground control system for UAS safe landing area determination (SLAD) in urban air mobility operations. *Sensors* **2022**, *22*, 3226. [CrossRef] [PubMed]

97. Zang, X.; Jiang, L.; Liang, C.; Dong, J.; Lu, W.; Mladenovic, N. Optimization approaches for the urban delivery problem with trucks and drones. *Swarm Evol. Comput.* **2022**, *75*, 101147. [CrossRef]

98. Resat, H.G. Design and analysis of novel hybrid multi-objective optimization approach for data-driven sustainable delivery systems. *IEEE Access* **2020**, *8*, 90280–90293. [CrossRef]

99. Brunner, G.; Szebedy, B.; Tanner, S.; Wattenhofer, R. The urban last mile problem: Autonomous drone delivery to your balcony. In Proceedings of the 2019 International Conference on Unmanned Aircraft Systems, ICUAS 2019, Atlanta, GA, USA, 11–14 June 2019; pp. 1005–1012. [CrossRef]

100. Bahabry, A.; Wan, X.; Ghazzai, H.; Menouar, H.; Vesonder, G.; Massoud, Y. Low-altitude navigation for multi-rotor drones in urban areas. *IEEE Access* **2019**, *7*, 87716–87731. [CrossRef]

101. Mayalu, A.; Kochersberger, K.; Jenkins, B.; Malassenet, F. Lidar data reduction for unmanned systems navigation in urban Canyon. *Remote. Sens.* **2020**, *12*, 1724. [CrossRef]

102. Li, A.; Hansen, M.; Zou, B. Traffic management and resource allocation for UAV-based parcel delivery in low-altitude urban space. *Transp. Res. Part C Emerg. Technol.* **2022**, *143*, 103808. [CrossRef]

103. Kuru, K. Planning the future of smart cities with swarms of fully autonomous unmanned aerial vehicles using a novel framework. *IEEE Access* **2021**, *9*, 6571–6595. [CrossRef]

104. Kirschstein, T. Energy demand of parcel delivery services with a mixed fleet of electric vehicles. *Clean. Eng. Technol.* **2021**, *5*, 100322. [CrossRef]

105. Aiello, G.; Inguanta, R.; D'Angelo, G.; Venticinque, M. Energy consumption model of aerial urban logistic infrastructures. *Energies* **2021**, *14*, 5998. [CrossRef]

106. Hong, I.; Kuby, M.; Murray, A.T. A range-restricted recharging station coverage model for drone delivery service planning. *Transp. Res. Part C Emerg. Technol.* **2018**, *90*, 198–212. [CrossRef]

107. Buko, J.; Bulsa, M.; Makowski, A. Spatial premises and key conditions for the use of UAVs for delivery of items on the example of the polish courier and postal services market. *Energies* **2022**, *15*, 1403. [CrossRef]

108. Sah, B.; Gupta, R.; Bani-Hani, D. Analysis of barriers to implement drone logistics. *Int. J. Logist. Res. Appl.* **2021**, *24*, 531–550. [CrossRef]

109. Cohen, A.P.; Shaheen, S.A.; Farrar, E.M. Urban air mobility: History, ecosystem, market potential, and challenges. *IEEE Trans. Intell. Transp. Syst.* **2021**, *22*, 6074–6087. [CrossRef]

110. Yoo, W.; Yu, E.; Jung, J. Drone delivery: Factors affecting the public's attitude and intention to adopt. *Telemat. Inform.* **2018**, *35*, 1687–1700. [CrossRef]

111. Kwon, D.; Son, S.; Park, Y.; Kim, H.; Park, Y.; Lee, S.; Jeon, Y. Design of secure handover authentication scheme for urban air mobility environments. *IEEE Access* **2022**, *10*, 42529–42541. [CrossRef]

112. Jaller, M.; Otero-Palencia, C.; Pahwa, A. Automation, electrification, and shared mobility in urban freight: Opportunities and challenges. *Transp. Res. Procedia* **2020**, *46*, 13–20. [CrossRef]

113. Han, P.; Yang, X.; Zhao, Y.; Guan, X.; Wang, S. Quantitative ground risk assessment for urban logistical unmanned aerial vehicle (UAV) based on bayesian network. *Sustainability* **2022**, *14*, 5733. [CrossRef]

114. Anbaroğlu, B. Parcel delivery in an urban environment using unmanned aerial systems: A vision paper. *ISPRS Ann. Photogramm. Remote. Sens. Spat. Inf. Sci.* **2017**, *4*, 73–79. [CrossRef]

115. Finn, R.L.; Wright, D. Unmanned aircraft systems: Surveillance, ethics and privacy in civil applications. *Comput. Law Secur. Rev.* **2012**, *28*, 184–194. [CrossRef]

116. Pedersen, C.B.; Rosenkrands, K.; Sung, I.; Nielsen, P. Systemic performance analysis on zoning for unmanned aerial vehicle-based service delivery. *Drones* **2022**, *6*, 157. [CrossRef]
117. Ahmed, S.S.; Hulme, K.F.; Fountas, G.; Eker, U.; Benedyk, I.V.; Still, S.E.; Anastasopoulos, P.C. The flying car—challenges and strategies toward future adoption. *Front. Built Environ.* **2020**, *6*, 1–11. [CrossRef]
118. Park, H.J.; Mirjalili, R.; Côté, M.J.; Lim, G.J. Scheduling diagnostic testing kit deliveries with the mothership and drone routing problem. *J. Intell. Robot. Syst. Theory Appl.* **2022**, *105*, 38. [CrossRef] [PubMed]
119. Kapoor, R.; Kloet, N.; Gardi, A.; Mohamed, A.; Sabatini, R. Sound propagation modelling for manned and unmanned aircraft noise assessment and mitigation: A review. *Atmosphere* **2021**, *12*, 1424. [CrossRef]
120. Torija, A.J.; Clark, C. A psychoacoustic approach to building knowledge about human response to noise of unmanned aerial vehicles. *Int. J. Environ. Res. Public Health* **2021**, *18*, 682. [CrossRef]
121. Stolaroff, J.K.; Samaras, C.; O'Neill, E.R.; Lubers, A.; Mitchell, A.S.; Ceperley, D. Energy use and life cycle greenhouse gas emissions of drones for commercial package delivery. *Nat. Commun.* **2018**, *9*, 409. [CrossRef]
122. Torija, A.J.; Li, Z.; Self, R.H. Effects of a hovering unmanned aerial vehicle on urban soundscapes perception. *Transp. Res. Part D Transp. Environ.* **2020**, *78*, 102195. [CrossRef]

