# Peer review of "Drone-Aided Delivery Methods, Challenge, and the Future: A Methodological Review"

_drones, doi:10.3390/drones7030191_

Round 1

Reviewer 1 Report

The work is interesting. it aims to review the current literature on drone delivery and identify research trends, challenges, and future research directions.

Author Response

Please see the attached response to comments.

Reviewer 2 Report

1. This paper provides a comprehensive overview of the latest research on drone delivery. However, it is necessary to conduct a more in-depth analysis, in order to discuss the challenges and limitations associated with drone delivery. Additionally, it would be valuable to consider the challenges faced by different stakeholders, including businesses, consumers, and regulators. A broader examination of the societal and economic implications of drone delivery could provide valuable insights for future research and implementation.

2. It is advisable to remove the years 2014 and 2023 from the axis in Figure 2 as there is no data available for these years, which might lead to ambiguity that there were no relevant articles published in 2014.

3. Please add a header in the first row of tables 1, 2, 4, and 5.

4. In the Abstract, it is stated "We then categorize the literature according to the characteristics and objectives of the problems and thoroughly analyze them based on mathematical formulations and solution techniques." However, there are no specific mathematical formulations included in the paper. Can you please provide further elaboration on this aspect?

5. It is suggested to provide simple explanations and illustrations for the concepts such as TSP, VRP, DDSP, DOP, and ULM in Section 3, in order to facilitate a more intuitive understanding of the paper for readers who are not familiar with the field.

6. It would be beneficial to analyze the advantages and disadvantages as well as the level of advancement of each of the five categories of drone delivery literature presented in detail, and to provide a creative and accurate conclusion.

7. In the conclusion, the key point of the statement is not prominent, it may be divided into sub headings. Additionally, the conclusion mostly summarizes the research methods used in this paper but lacks sufficient valuable analysis. It is hoped that this aspect can be supplemented.

8. The novelty of the research method is not enough, the method stated in Figure 1 is basically the method used by each researcher to read related literature, and the overall writing content is more like the stacking of reference literature content.

9. The article may be more persuasive if it includes relevant data to support, even though it is a methodology study.

10. There are some grammar mistakes throughout the paper. Please proofread the manuscript and do multiple checks.

1) L36 various applications

2) L52 less fewer mathematical modeling details

3) L247 Others Other contributions

4) L268 Xia et al. [63] was focused on

5) L286 in others other cases

6) The expression in the middle column of the last two rows in Table 2, "Stochastic modeling" and "Stochastic modelling", is inconsistent.

Reviewer 3 Report

The study deserves publication but could be improved.

The paper is addressing a multitude differing publications on various topics of logistical models. As it is, it is addressing only part of the research universe related drones and mainly parcel delivery and is obviously mostly interesting for people working with these specific topics. That is not a serious problem, but there is a large segment of drone applications related to other purposes, this should at least be mentioned and briefly discussed. In particular, it would be interesting if they discussed drone solutions that combine routing problems with on-demand services. What models are most appropriate in such services?

After reading the paper, I am left with brief statements from a variety of studies and ask myself; are there any major learning? May I learn any association between the type of model and the character of a given service need? Although it is a methodological review, this would offer added value for many readers. Bu may be I missed something here.

Method.

The authors have applied a methodological review which is expected to focus on summarizing the state-of-the-art methodological practices of the topic they have studied. Their search strategy resulted in 165 articles (surprisingly low to me), reduced to inclusion of 73 articles. Was this obtained from processing the 165, or does it include the snowballing papers? Criteria for selection/supplementing snowballing papers is not fully described, and discussion of possible biases should be included.

Content.

They authors advertise that they intend to cover a gap in the current literature. What gap? Based on their search criteria they have studied relevant literature of the topics they have advertised, but they have obviously not harvested a lot of literature which is closely related to their topics. I do not consider this as a problem because they define their method, but more as a fact that could be better discussed or described.

There is a vast literature related to the logistic models that the authors discussed. As I observe it from my interest of drone research, much of this literature has been sparked by the fact that drones currently have short ranges, limited weight capacities and are rather susceptible to weather conditions, thus introducing the aspect of short flights, frequent charging stations and so on. I wonder how many of these models are relevant for drone services when drones overtime will gain more robustness, longer range and increased weight capacities? That is crucial for future research. I would find it very valuable to include such a discussions in this paper.

They announce that they will identify directions for future research. This is rather marginally in the paper, and should definitely be discussed more in depth.

This could address two different perspectives. Whether the current models also will apply for future drone solutions of longer range when drones are becoming more robust and powerful, or whether some of the current literature will be kind of outdated as drone performance strengthen. I am convinced that the current need of frequent battery changes and charging will be eliminated in a few years. If I’m right, what crucial topics are we then left with? Timing? Costs? I think this is an interesting perspective which the authors should include.

Actually, I do not find any interesting discussion or reflections with substance that might enlighten me on how I should plan my future research topics related to mathematical models for logistics. Some discussion of this would improve the paper significantly.

When the authors present the results and describe the different studies included, it ends up as a rather challenging part, giving varying, more or less detailed descriptions of the different previous studies, but scare comments/comparisons related to weaknesses/strengths. They then summarize in table format. The authors state that the overall objective of this study is to provide a critical analysis of the methodologies. This “critical aspect” is not easy to access as a reader. May be a richer description in the table format could make this part of the paper more reader friendly?

Conclusion & Limitations.

This chapter mostly tell what the authors did in the study. We have observed that already. I would rather like to use this space for some of the aspects I have tried to illustrate above related to future research.

Round 2

Reviewer 2 Report

Accept in present form